# Data-Free One-Shot Federated Learning Under Very High Statistical Heterogeneity

**Clare Heinbaugh**[*]**, Emilio Luz-Ricca**[*]**, Huajie Shao**[†]
Department of Computer Science, William & Mary, VA, USA
{ceheinbaugh, eluzricca, hshao}@wm.edu
[*]Equal contribution, [†]Corresponding author

## Abstract

Federated learning (FL) is an emerging distributed learning framework that collaboratively trains a shared model without transferring the local clients' data to a centralized server. Motivated by concerns stemming from extended communication and potential attacks, one-shot FL limits communication to a single round while attempting to retain performance. However, one-shot FL methods often degrade under high statistical heterogeneity, fail to promote pipeline security, or require an auxiliary public dataset. To address these limitations, we propose two novel data-free one-shot FL methods: FEDCVAE-ENS and its extension FEDCVAE-KD. Both approaches reframe the local learning task using a conditional variational autoencoder (CVAE) to address high statistical heterogeneity. Furthermore, FEDCVAE-KD leverages knowledge distillation to compress the ensemble of client decoders into a single decoder. We propose a method that shifts the center of the CVAE prior distribution and experimentally demonstrate that this promotes security, and show how either method can incorporate heterogeneous local models. We confirm the efficacy of the proposed methods over baselines under high statistical heterogeneity using multiple benchmark datasets. In particular, at the highest levels of statistical heterogeneity, both FEDCVAE-ENS and FEDCVAE-KD typically more than double the accuracy of the baselines.

## 1 Introduction

Traditional federated learning (FL) achieves privacy protection by sharing learned model parameters with a central server, circumventing the need for a centralized dataset and thus allowing potentially sensitive data to remain local to client devices (McMahan et al., 2017). FL has shown promise in several practical application domains with privacy concerns, such as health care, mobile phones, and industrial engineering (Li et al., 2020a). However, most existing FL methods depend on substantial iterative communication (Guha et al., 2019; Li et al., 2020b), introducing a vulnerability to eavesdropping attacks, among other privacy and security concerns (Mothukuri et al., 2021).

One-shot FL has emerged to address issues associated with communication and security in standard FL (Guha et al., 2019). One-shot FL limits communication to a single round, which is more practical in scenarios like model markets, where models trained to convergence are sold with no possibility for iterative communication during local client training (Li et al., 2021b). In high impact settings, like health care, data could be highly heterogeneous and computation capabilities could be varied; for example, health care institutions could have different prevalence rates of particular diseases or no data on a disease and substantially different computing abilities depending on funding (Li et al., 2020a). Furthermore, fewer communications rounds means fewer opportunities for eavesdropping attacks. While results in one-shot FL are promising, existing methods struggle under high statistical heterogeneity, non-independently- and identically-distributed (non-IID) data, (i.e., Zhou et al. (2020) Zhang et al. (2021)) or do not fully consider statistical heterogeneity (i.e., Guha et al. (2019), Shin et al. (2020), Li et al. (2021b)). Additionally, most do not consider pipeline security (i.e., Shin et al. (2020), Li et al. (2021b), Zhang et al. (2021)). Furthermore, an auxiliary public dataset is often required to achieve satisfactory performance in one-shot FL (i.e., Guha et al. (2019), Li et al. (2021b)), which may be difficult to obtain in practice (Zhu et al., 2021).

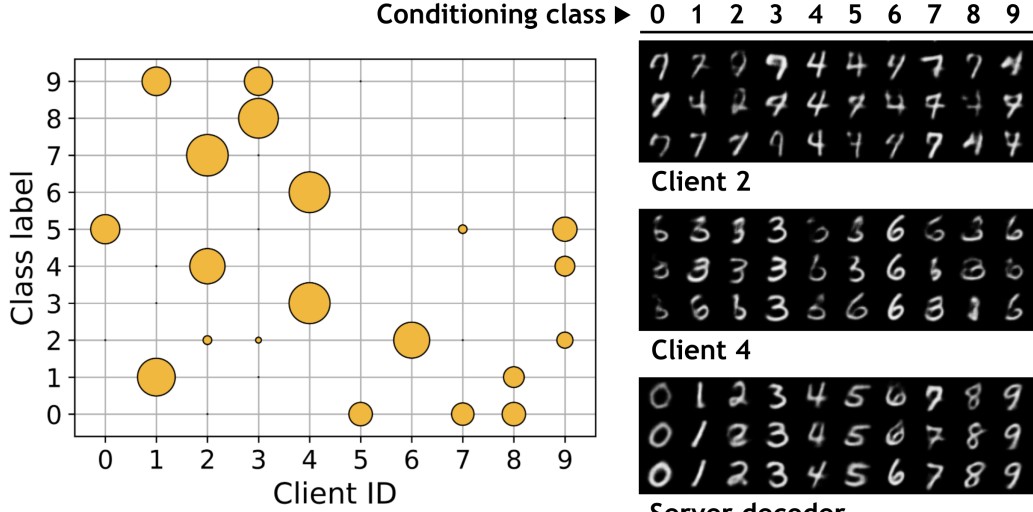

Figure 1: Motivating our proposed methods, FEDCVAE-ENS and FEDCVAE-KD, using the MNIST dataset as an example. In cases of very high statistical heterogeneity, each client will only observe one or two of the ten available classes, as seen on the left where the size of each dot is proportional to the number of samples. For example, client 2 only observes 4's and 7's, resulting in a client decoder that can expertly generate these digits. Note that the columns are shown in order of conditioning class (digits 0-9). Similarly, client 4 is an expert in 3's and 6's. In FEDCVAE-KD, our lightweight knowledge distillation training procedure compacts local learning into a single server decoder, as evidenced by the high-quality samples from all available classes (digits 0-9). This server decoder can then be used for any downstream task, e.g., classification.

To address these issues, we jointly propose FEDCVAE-ENS and FEDCVAE-KD, two novel data-free one-shot FL models that reframe the local learning task using conditional variational autoencoders (CVAE). Because CVAEs can easily learn a simplified data distribution, both methods train CVAEs locally to capture the narrow conditional data distributions that arise in the high statistical heterogeneity setting. Figure 1 shows how client decoders become experts in the few classes that they observed. These decoders are ensembled (FEDCVAE-ENS) or compactly aggregated (FEDCVAE-KD). More specifically, FEDCVAE-KD aggregates the models using a lightweight knowledge distillation procedure; client decoders are teachers, and the server decoder is the student. Figure 1 shows images generated by the server decoder.

Thorough experiments on multiple benchmark datasets (MNIST, FashionMNIST, SVHN) demonstrate the superiority of FEDCVAE-ENS and FEDCVAE-KD over other relevant one-shot FL methods in the high statistical heterogeneity setting. In particular, FEDCVAE-ENS and FEDCVAE-KD obtain more than $1.75\times$ the accuracy of the best baseline method for MNIST, more than $2\times$ the accuracy for FashionMNIST, and more than $2.75\times$ the accuracy for SVHN under extreme statistical heterogeneity (i.e., clients only observe one or two classes). Furthermore, to protect the decoders uploaded to the server, we propose a method to shift the center of the CVAE prior distribution. We show that without knowing the center of the prior, an eavesdropping attacker cannot train a performant classifier, thus promoting pipeline security.

In sum, our contributions are two one-shot FL methods targeted to the high statistical heterogeneity setting that: (1) perform substantially better than other baseline methods in this setting, (2) demonstrate invariance to the number of clients, (3) are data-free and can be applied to any downstream task requiring a labeled dataset, (4) allow for heterogeneous local model architectures, and (5) extend to promote pipeline security. To the best of our knowledge, we are the first to thoroughly address very high statistical heterogeneity in one-shot FL.

## 2 PRELIMINARIES

**Conditional Variational Autoencoders**. A variational autoencoder (VAE) is a probabilistic generative model that attempts to learn the distribution of data samples (Kingma & Welling, 2014).

A VAE is a latent variable method that models the joint distribution $p_{\boldsymbol{\theta}}(\mathbf{x}, \mathbf{z})$ of a data sample $\mathbf{x} \in \mathcal{X} \subseteq \mathbb{R}^D$ and latent variable $\mathbf{z} \in \mathcal{Z} \subseteq \mathbb{R}^d$, where usually $d << D$. This joint can be factorized as $p_{\boldsymbol{\theta}}(\mathbf{x}, \mathbf{z}) = p_{\boldsymbol{\theta}}(\mathbf{x}|\mathbf{z})p(\mathbf{z})$, where the prior $p(\mathbf{z})$ is usually chosen to be a multivariate standard normal distribution, i.e., $p(\mathbf{z}) = \mathcal{N}(\mathbf{0}, \mathbf{I})$. The posterior $p_{\boldsymbol{\theta}}(\mathbf{z}|\mathbf{x})$ is approximated via an inference model $q_{\boldsymbol{\phi}}(\mathbf{z}|\mathbf{x})$ called the *encoder* and the model for the conditional likelihood $p_{\boldsymbol{\theta}}(\mathbf{x}|\mathbf{z})$ is called the *decoder*. In our case, both the encoder and decoder are deep neural networks parameterized by $\boldsymbol{\phi}$ and $\boldsymbol{\theta}$, respectively. Conditional VAEs (CVAEs) extend the basic VAE by conditioning the encoder and decoder on the one-hot encoding of the class label $\mathbf{y} \in \mathcal{Y} = \{1, 2, ..., K\}$ corresponding to data sample $\mathbf{x}$, resulting in the conditional encoder $q_{\boldsymbol{\phi}}(\mathbf{z}|\mathbf{x}, \mathbf{y})$ and conditional decoder $p_{\boldsymbol{\theta}}(\mathbf{x}|\mathbf{z}, \mathbf{y})$. A CVAE is trained by maximizing the variational lower bound:

$$\mathcal{L}(\boldsymbol{\phi}, \boldsymbol{\theta}; \mathbf{x}, \mathbf{y}) = -D_{\text{KL}}(q_{\boldsymbol{\phi}}(\mathbf{z}|\mathbf{x}, \mathbf{y}) \,||\, p(\mathbf{z})) + \mathbb{E}_{q_{\boldsymbol{\phi}}(\mathbf{z}|\mathbf{x}, \mathbf{y})}[\log p_{\boldsymbol{\theta}}(\mathbf{x}|\mathbf{z}, \mathbf{y})], \quad (1)$$

which bounds the conditional marginal likelihood of the data $\log p(\mathbf{x}|\mathbf{y})$. Here, $D_{\text{KL}}(\cdot)$ represents KL-divergence.

**Knowledge Distillation.** Knowledge distillation (KD) aims to extract information from a trained *teacher* model to train a separate lightweight *student* model (Buciluǎ et al., 2006; Hinton et al., 2015). Typically, the student model is trained by minimizing the discrepancy between student and teacher logits generated using a suitable auxiliary dataset (Hinton et al., 2015); KL-divergence is often chosen as the measure of discrepancy. Some works ensemble teacher models, using the average of teacher logits in an attempt to compact ensemble knowledge into a single student model (Anil et al., 2018; Dvornik et al., 2019; Furlanello et al., 2018). Several works have integrated KD into FL to mitigate privacy risks, reduce upload costs, or regularize local learning using an ensemble-of-teachers approach (Lin et al., 2020; Zhu et al., 2021; Guha et al., 2019). However, KD approaches in FL usually require an auxiliary public dataset with similar properties as the distributed dataset, which may be difficult to obtain in practice (Zhu et al., 2021).

**One-shot Federated Learning.** In the federated setting, we have a set of clients $\mathcal{C}$, with $m = |\mathcal{C}|$ clients in total. Each client $k$ has a local private dataset $\mathcal{D}^k = \{(\mathbf{x}_i, \mathbf{y}_i)\}_{i=1}^{n_k}$, with $n_k = |\mathcal{D}^k|$ representing the number of data samples belonging to user $k$. Traditional FL methods assume each client has a local differentiable model $f_{\mathbf{w}^k}(\cdot)$, usually a deep neural network parameterized by $\mathbf{w}^k$. It is typically assumed that the server and clients can communicate over multiple rounds. However, in the one-shot FL setting communication is restricted to a single round, which severely limits communication costs but also increases the difficulty of the distributed learning task (Guha et al., 2019). Notably, existing one-shot FL methods either ignore the issue of statistical heterogeneity (i.e., Guha et al. (2019)), fail to comprehensively explore the effect of statistical heterogeneity on performance (i.e., Shin et al. (2020) and Li et al. (2021b)), or degrade substantially at even moderate levels of statistical heterogeneity (i.e., Zhou et al. (2020) and Zhang et al. (2021)).

## 3 TACKLING VERY HIGH STATISTICAL HETEROGENEITY IN ONE-SHOT FL

We jointly propose FEDCVAE-ENS and FEDCVAE-KD, one-shot FL methods that do not require an auxiliary public dataset for server-side training (they are *data-free*). They address issues caused by high statistical heterogeneity by reframing the learning task using CVAEs and account for model heterogeneity by allowing different CVAE architectures across clients. FEDCVAE-ENS is described in Algorithm 2 (Appendix A) and visualized in Figure 6 (Appendix A), with FEDCVAE-KD described in Algorithm 1 and visualized in Figure 2. We discuss privacy- and security-promoting extensions in Appendix B and Section 3.3, respectively.

### 3.1 OVERVIEW

Figure 2 illustrates the overall framework of the proposed one-shot method FEDCVAE-KD and Figure 6 (Appendix A) shows FEDCVAE-ENS. Specifically, clients first train CVAEs locally on their private data. Next, each client's trained decoder parameters and local label distributions are uploaded to the server once; *this is the only communication round*. Then, the server generates samples from the client decoders according to the client's local label distribution. Generating samples based on a client's label distribution ensures that each client presents samples from the classes that they know best. In FEDCVAE-ENS, these generated samples are directly used to perform a downstream task, e.g., train a classifier. In FEDCVAE-KD, these samples are used to train a single server

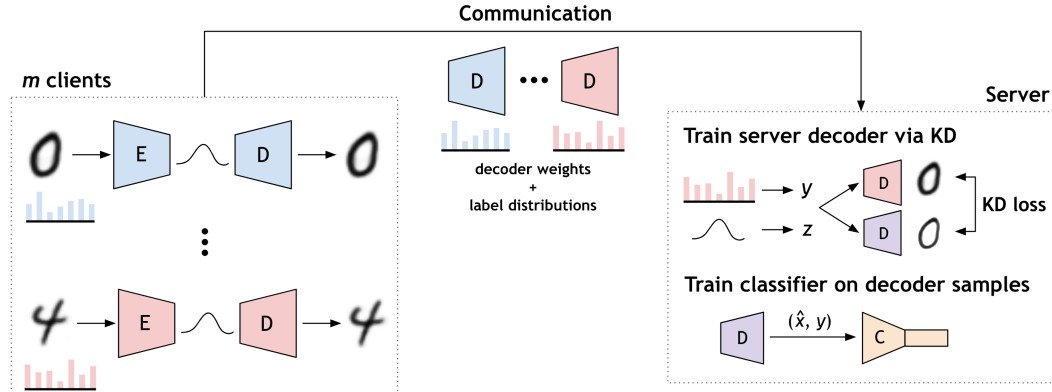

Figure 2: The full pipeline for one of our proposed methods, FEDCVAE-KD. Here, E, D, and C represent "encoder," "decoder," and "classifier" models, respectively. First, clients train CVAEs on their private local datasets. Then, the server uses uploaded client decoder parameters and local label distributions to train a server decoder using knowledge distillation (KD). Finally, synthetic labeled samples from the server decoder are used to train a classifier.

decoder via KD. Thus, the single conditional decoder can be used as a compact labeled dataset to perform a task like training a classifier, as depicted in Figure 2. Because FEDCVAE-KD extends FEDCVAE-ENS, we leave the description of FEDCVAE-ENS to Appendix A.

## 3.2 FEDCVAE-KD: DECODER AGGREGATION USING KNOWLEDGE DISTILLATION

FEDCVAE-KD trains a CVAE $f_{\mathbf{w}^k}(\cdot)$ for every client $k \in \mathcal{C}$ to convergence on their private local dataset by solving Equation 1; this CVAE is parameterized by $\mathbf{w}^k = [\boldsymbol{\phi}^k, \boldsymbol{\theta}^k]$, with encoder $E_{\boldsymbol{\phi}^k}(\cdot)$ and decoder $D_{\boldsymbol{\theta}^k}(\cdot)$. Then, clients communicate their decoder weights $\boldsymbol{\theta}^k$ and label distributions $\hat{p}^k(\mathbf{y})$ to the server; in practice, this simply requires upload of client label counts as in Zhu et al. (2021). This completes the single communication round.

Now we move to the server. Intuitively, if a CVAE observes samples primarily from only a few of the $K$ total classes (as is likely when data is highly heterogeneous), the CVAE will become an "expert" in the simplified data distribution over those few classes. To ensure each client only presents its highest-quality samples, we generate conditioning classes $\mathbf{y}$ by sampling from the client's local label distribution, i.e., $\mathbf{y} \sim \hat{p}^k(\mathbf{y})$. Then, to sample from client decoders, we sample a latent vector from the prior (i.e., $\mathbf{z} \sim \mathcal{N}(\mathbf{0}, \mathbf{I})$)[1] and obtain synthetic data sample $i$ from client $k$ using $\hat{\mathbf{x}}_i^k = D_{\boldsymbol{\theta}^k}(\mathbf{z}_i^k; \mathbf{y}_i^k)$.

The trained client decoders act as the teacher models, conveying their aggregate knowledge of how to map from latent space to data space to a single student server decoder, parameterized by $\boldsymbol{\theta}^S$. To achieve this, we generate $n_D$ total KD training samples, with $\mathcal{D}_{\text{Ens}}$ defined as the combination of client subsets $\mathcal{D}_{\text{Ens}}^k = \{(\hat{\mathbf{x}}_i^k, \mathbf{y}_i^k, \mathbf{z}_i^k)\}_{i=1}^{\lfloor n_D/m \rfloor}$. Then, we train the server to match the teacher's mapping of $(\mathbf{z}_i^k, \mathbf{y}_i^k)$ to $\hat{\mathbf{x}}_i^k$ by minimizing a reconstruction loss:

$$\ell_{\text{KD}}(\boldsymbol{\theta}^S; \mathbf{z}^k, \mathbf{y}^k, \hat{\mathbf{x}}^k) = g(D_{\boldsymbol{\theta}^S}(\mathbf{z}^k; \mathbf{y}^k), \hat{\mathbf{x}}^k), \tag{2}$$

which penalizes the dissimilarity $g(\cdot)$ in data space between the synthetic data sample generated by the server decoder $D_{\boldsymbol{\theta}^S}(\mathbf{z}^k; \mathbf{y}^k)$ and the client decoder sample $\hat{\mathbf{x}}^k$. To facilitate comparison with existing works in one-shot FL, we use the trained server decoder to generate an IID labeled dataset $\mathcal{D}_C$ of $n_C$ samples to train the server classifier $f_{\mathbf{w}_C^S}(\cdot)$, parameterized by $\mathbf{w}_C^S$.[2]

---

[1]In practice, we find it useful to focus on the highest density region of the prior and instead sample from a truncated standard normal distribution with tight symmetric bounds.

[2]If classification is the downstream task, we note that rather than train an auxiliary classifier in the server, the server decoder's conditional likelihood model $p(\mathbf{x}|\mathbf{z}, \mathbf{y})$ could be used directly in the generative classifier $p(\mathbf{y}|\mathbf{x}) = \frac{\int p(\mathbf{x}|\mathbf{y}, \mathbf{z})p(\mathbf{z})d\mathbf{z} \cdot p(\mathbf{y})}{p(\mathbf{x})}$. We leave further exploration of this modeling direction to future work.

---

**Algorithm 1 - FEDCVAE-KD in the one-shot FL setting.** $T_L$ represents the number of local training epochs. The server decoder parameters are $\boldsymbol{\theta}^S$, with KD training epochs $T_{\text{KD}}$, number of KD training samples $n_D$, KD loss $\ell_{\text{KD}}(\cdot)$, and KD learning rate $\eta_{\text{KD}}$. The server classifier parameters are $\mathbf{w}_C^S$, with training epochs $T_C$, number of training samples $n_C$, classification loss $\ell_C(\cdot)$, and learning rate $\eta_C$. $\mathcal{C}$ is the set of clients.

---

1: **procedure** SERVER
2:     **for** each client $k \in \mathcal{C}$ **in parallel do**
3:         $\boldsymbol{\theta}^k$, $\hat{p}^k(\mathbf{y}) \leftarrow$ **ClientLocalUpdate**$(k, T_L)$           $\triangleright$ See Algorithm 3 in Appendix A
4:     Generate samples from client $\mathcal{D}_{\text{Ens}}^k := \{(\hat{\mathbf{x}}_i^k, \mathbf{y}_i^k, \mathbf{z}_i^k)\}_{i=1}^{\lfloor n_D/m \rfloor}$ using client decoder $D_{\boldsymbol{\theta}^k}(\cdot)$
        and label distribution $\hat{p}^k(\mathbf{y})$
5:     Combine client subsets into a KD dataset $\mathcal{D}_{\text{Ens}} := \mathcal{D}_{\text{Ens}}^1 \cup \mathcal{D}_{\text{Ens}}^2 \cup ... \cup \mathcal{D}_{\text{Ens}}^m$
6:     **for** server decoder epoch $i = 1$ to $T_{\text{KD}}$ **do**
7:         **for** mini-batch $b \subset \mathcal{D}_{\text{Ens}}$ **do**
8:             $\boldsymbol{\theta}^S \leftarrow \boldsymbol{\theta}^S - \eta_{\text{KD}} \cdot \nabla_{\boldsymbol{\theta}^S} \ell_{\text{KD}}(\boldsymbol{\theta}^S; b)$
9:     Generate an IID labeled dataset $\mathcal{D}_C := \{(\hat{\mathbf{x}}_i^S, \mathbf{y}_i^S)\}_{i=1}^{n_C}$, by sampling from trained server
        decoder $D_{\boldsymbol{\theta}^S}(\cdot)$
10:     **for** classifier epoch $i = 1$ to $T_C$ **do**
11:         **for** mini-batch $b \subset \mathcal{D}_C$ **do**
12:             $\mathbf{w}_C^S \leftarrow \mathbf{w}_C^S - \eta_C \cdot \nabla_{\mathbf{w}_C^S} \ell_C(\mathbf{w}_C^S; b)$

---

Because FEDCVAE-KD ensembles client decoders to create a labeled dataset $\mathcal{D}_{\text{Ens}}$ to train the server decoder, each client can have a unique CVAE model architecture, accommodating each client's computational limitations. Furthermore, the decision of the classifier architecture can be deferred until after FL is finished and will not affect the learning procedure. FEDCVAE-KD can be applied to any task that requires a labeled dataset, which is more general than classification; there is no commitment to a particular terminal task before learning occurs. While we do not explore the extended communication setting, we note that FEDCVAE-KD extends naturally by communicating the server decoder parameters obtained through KD to all clients and repeating the outlined procedure for non-terminal communication rounds.

### 3.3 SECURITY-PROMOTING EXTENSION

We define a secure pipeline as one where an outside attacker who obtains transferred data cannot train a performant classifier (Zhou et al., 2020). In the case of FEDCVAE-ENS and FEDCVAE-KD, an attacker who intercepts all client decoders and local label distributions should not be able to generate the high quality samples necessary to train a high quality classifier.

CVAEs use a prior distribution over latent space to train the encoder and decoder models. While a multivariate standard normal distribution is typically used for convenience, any normal distribution is acceptable. To promote security, we propose to shift the center of the prior distribution $\boldsymbol{\mu}$ to a random position in real space (i.e., $\boldsymbol{\mu} \in \mathbb{R}^d$), which can be communicated offline or via encryption methods between server and clients (Zhou et al., 2020). As shown in Figure 7 (Appendix B), sampling latent vectors too far from the center of the normal prior produces qualitatively poor data samples, deterring eavesdropping attackers who have no knowledge of $\boldsymbol{\mu}$. We conduct experiments to verify the effectiveness of this extension.

## 4 EXPERIMENTAL EVALUATION

### 4.1 SETUP

**Benchmark Datasets.** To validate FEDCVAE-ENS and FEDCVAE-KD, we conduct experiments on three image datasets that are standard in the FL literature: MNIST (Lecun et al., 1998), Fashion-MNIST (Xiao et al., 2017), and SVHN (Netzer et al., 2011).[3] Datasets are described in Appendix

---

[3]The code used to implement our proposed methods and carry out all experiments is included in the following public repository: https://github.com/ceh-2000/fed_cvae.

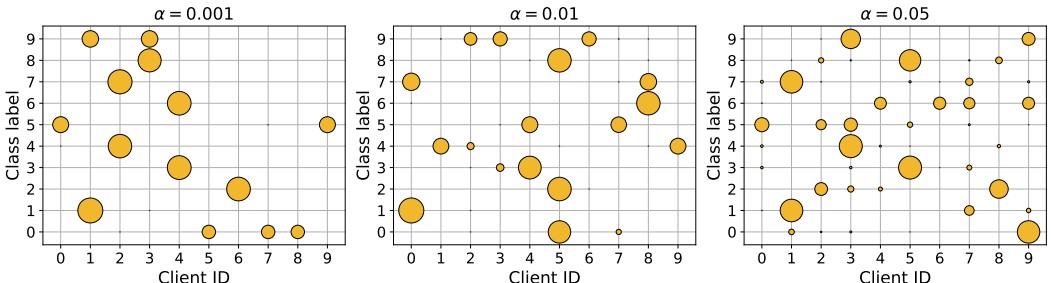

Figure 3: Example distributions of class labels for MNIST with $m = 10$ clients over multiple levels of statistical heterogeneity $\alpha$. The size of each dot is proportional to the number of samples.

C. To simulate statistical heterogeneity, we use the Dirichlet distribution to generate disjoint non-IID client training datasets as in Hsu et al. (2019) and Lin et al. (2020). Specifically, we sample $\mathbf{p}^k \sim Dir(\alpha)$ and allocate a $\mathbf{p}_i^k$ proportion of class $i$ to client $k$. The parameter $\alpha$ controls the level of non-IID-ness, with a lower $\alpha$ inducing more skewed label distributions across clients. To illustrate the effect of $\alpha$ on dataset partitions across clients, we visualize the distribution of labels across $m = 10$ clients for $\alpha = \{0.001, 0.01, 0.05\}$ in Figure 3.

**Baseline Methods.** We compare the performance of FEDCVAE-ENS and FEDCVAE-KD in the one-shot data-free FL setting against two existing methods: FEDAVG (McMahan et al., 2017) and a method proposed in Guha et al. (2019), which we call FEDONESHOT. FEDONESHOT ensembles the predictions of select uploaded client classifiers using a sampling procedure; because we consider substantially less clients than Guha et al. (2019), we disregard sampling and use all clients in the ensemble. There are recent FL methods that are not appropriate in our proposed setting. The one-shot methods proposed in Li et al. (2021b) and Shin et al. (2020) are not applicable because of their reliance on public auxiliary data for server-side training or fine-tuning. Similarly, many standard FL methods are not appropriate because they depend on an auxiliary dataset (i.e., FEDDF (Lin et al., 2020)) or focus on regularization (i.e., FEDGEN (Zhu et al., 2021), FEDPROX (Li et al., 2020c), SCAFFOLD (Karimireddy et al., 2020), FEDNOVA (Wang et al., 2020)), which is incompatible with the one-shot setting.

**Configurations.** Unless otherwise stated, we use $m = 10$ clients, $\alpha = 0.01$ (very heterogeneous), and report average test accuracy across 5 seeded parameter intializations $\pm$ one standard deviation. The data partition is fixed unless otherwise stated. Following Zhu et al. (2021), we distribute $50\%$ of the available training data to clients for MNIST and FashionMNIST, and $100\%$ for SVHN. All available test data is used to evaluate the final server classifier (or ensemble for FEDONESHOT). We adopt the same convolutional classifier architecture as McMahan et al. (2017) for all methods. We base our CVAE architecture on Higgins et al. (2017). The server decoder for FEDCVAE-KD has the same architecture as the client CVAEs by default, although this is not strictly necessary because FEDCVAE-KD supports heterogeneous CVAE architectures.

For each method, hyperparameters were obtained through tuning, with the bounds of the search grid extended until the best-performing value appeared in the middle of the grid. Full hyperparameter settings can be found in Table 3 and Table 4 (Appendix C). All classifiers use a cross-entropy objective. CVAE training uses binary cross entropy and mean squared error for the reconstruction term of the objective for grayscale and RGB images, respectively; we use the same reconstruction objective for the KD loss ($g(\cdot)$ in Equation 2).

## 4.2 GENERAL RESULTS

**Statistical Heterogeneity.** To demonstrate the efficacy of FEDCVAE-ENS and FEDCVAE-KD in the difficult setting of high statistical heterogeneity, we test on varying levels of $\alpha$, from high ($\alpha = 0.05$), to very high ($\alpha = 0.01$), to extreme ($\alpha = 0.001$) statistical heterogeneity. FEDCVAE-ENS consistently outperforms all other methods, and FEDCVAE-KD outperforms the baselines in nearly all datasets and levels of $\alpha$ (the only exception is MNIST at the lowest level of statistical heterogeneity) as shown in Table 1. At $\alpha = 0.001$, FEDCVAE-KD obtains more than $1.75\times$ the

Table 1: Performance of four data-free one-shot FL methods over three datasets and across three levels of statistical heterogeneity (lower $\alpha$ is more heterogeneous). Best results for each dataset and each level of $\alpha$ are in purple, with second best results in yellow.

|  | Heterogeneity | FEDAVG | FEDONESHOT | FEDCVAE-KD (ours) | FEDCVAE-ENS (ours) |
|---|---|---|---|---|---|
| MNIST | $\alpha = 0.001$ | $45.12 \pm 5.87$ | $11.90 \pm 0.40$ | $82.24 \pm 1.09$ | $95.37 \pm 0.52$ |
|  | $\alpha = 0.01$ | $58.29 \pm 2.83$ | $41.15 \pm 0.70$ | $82.01 \pm 1.61$ | $93.83 \pm 1.53$ |
|  | $\alpha = 0.05$ | $80.10 \pm 2.35$ | $82.95 \pm 0.49$ | $79.57 \pm 1.17$ | $91.86 \pm 0.75$ |
| FashionMNIST | $\alpha = 0.001$ | $32.77 \pm 4.52$ | $10.00 \pm 0.00$ | $69.53 \pm 1.70$ | $76.04 \pm 0.93$ |
|  | $\alpha = 0.01$ | $45.85 \pm 2.95$ | $37.63 \pm 0.53$ | $69.97 \pm 1.63$ | $76.62 \pm 1.61$ |
|  | $\alpha = 0.05$ | $46.86 \pm 2.37$ | $64.53 \pm 1.73$ | $67.24 \pm 1.96$ | $71.95 \pm 2.17$ |
| SVHN | $\alpha = 0.001$ | $20.31 \pm 4.36$ | $15.94 \pm 0.00$ | $55.48 \pm 2.12$ | $65.52 \pm 0.66$ |
|  | $\alpha = 0.01$ | $25.12 \pm 2.07$ | $31.91 \pm 1.26$ | $55.97 \pm 0.38$ | $65.50 \pm 2.28$ |
|  | $\alpha = 0.05$ | $33.66 \pm 4.12$ | $49.60 \pm 1.59$ | $54.24 \pm 2.08$ | $66.61 \pm 2.09$ |

Table 2: Performance of four one-shot FL methods over three datasets and across four numbers of clients $m$. Best results for each dataset and each level of $m$ are in purple, with second best results in yellow.

|  | # of Clients | FEDAVG | FEDONESHOT | FEDCVAE-KD (ours) | FEDCVAE-ENS (ours) |
|---|---|---|---|---|---|
| MNIST | $m = 5$ | $42.34 \pm 1.75$ | $42.97 \pm 1.54$ | $80.19 \pm 1.76$ | $92.81 \pm 2.16$ |
|  | $m = 10$ | $58.29 \pm 2.83$ | $41.15 \pm 0.70$ | $82.01 \pm 1.61$ | $93.83 \pm 1.53$ |
|  | $m = 20$ | $37.14 \pm 3.87$ | $32.22 \pm 0.71$ | $81.93 \pm 1.75$ | $93.52 \pm 0.86$ |
|  | $m = 50$ | $28.26 \pm 7.98$ | $35.62 \pm 0.66$ | $77.68 \pm 1.39$ | $87.56 \pm 1.44$ |
| FashionMNIST | $m = 5$ | $38.51 \pm 2.40$ | $32.23 \pm 1.68$ | $68.01 \pm 1.56$ | $73.70 \pm 1.93$ |
|  | $m = 10$ | $45.85 \pm 2.95$ | $37.63 \pm 0.53$ | $69.97 \pm 1.63$ | $76.62 \pm 1.61$ |
|  | $m = 20$ | $27.49 \pm 4.77$ | $24.72 \pm 1.20$ | $71.28 \pm 1.17$ | $76.51 \pm 1.38$ |
|  | $m = 50$ | $31.91 \pm 2.20$ | $41.01 \pm 2.49$ | $69.25 \pm 1.13$ | $76.13 \pm 2.00$ |
| SVHN | $m = 5$ | $40.61 \pm 0.64$ | $42.05 \pm 1.31$ | $57.17 \pm 2.56$ | $65.72 \pm 2.17$ |
|  | $m = 10$ | $25.12 \pm 2.07$ | $31.91 \pm 1.26$ | $55.97 \pm 0.38$ | $65.50 \pm 2.28$ |
|  | $m = 20$ | $26.99 \pm 3.15$ | $10.68 \pm 0.70$ | $51.80 \pm 2.75$ | $65.54 \pm 1.89$ |
|  | $m = 50$ | $19.59 \pm 0.00$ | $30.20 \pm 0.64$ | $40.03 \pm 2.62$ | $64.81 \pm 2.44$ |

accuracy of the best baseline method for MNIST, more than $2\times$ the accuracy for FashionMNIST, and nearly $2.75\times$ the accuracy for SVHN (Table 1). While the baselines FEDAVG and FEDONESHOT are very sensitive to the level of statistical heterogeneity, both FEDCVAE-ENS and FEDCVAE-KD demonstrate consistent performance across levels of $\alpha$ (Table 1).

**Number of Clients.** Because FL applications often include many participating clients (Li et al., 2020a), we evaluate several values for number of clients ($m = \{5, 10, 20, 50\}$). While both FEDAVG and FEDONESHOT struggle when many clients are present, Table 2 shows that FEDCVAE-ENS and FEDCVAE-KD perform consistently across the number of clients, with the only exception being FEDCVAE-KD for SVHN. Even though it is typical to observe FL methods' accuracy degrade with increasing numbers of clients (Zhang et al., 2021), FEDAVG and FEDONESHOT are unstable with no clear decrease in accuracy, which we ascribe to the highly variable partitions generated at high levels of statistical heterogeneity. Experiments varying the dataset partitions reveal high variation in accuracy for FEDAVG and FEDONESHOT, whereas both FEDCVAE-ENS and FEDCVAE-KD exhibit consistent accuracy (Table 5 in Appendix C).

**Decoder Aggregation.** The proposed KD aggregation method used in FEDCVAE-KD substantially improves on aggregation via parameter averaging, generating qualitatively more realistic samples (Figure 8 in Appendix C) and achieving higher server classifier accuracy (Table 6 in Appendix C). As expected, FEDCVAE-ENS bounds the performance of FEDCVAE-KD, but KD-based aggregation still preserves client ensemble performance well across levels of $\alpha$ for all datasets (Table 1). The upper bound accuracy established by FEDCVAE-ENS is promising, but successful decoder aggregation may require a stronger KD procedure (e.g., feature-based KD as described in Gou et al. (2021)).

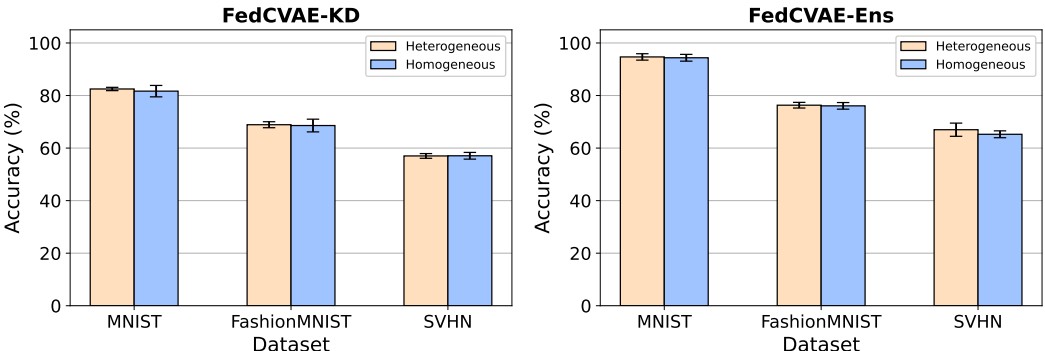

Figure 4: Results with heterogeneous local models. "Homogeneous" uses the same CVAE architecture for all clients, whereas "heterogeneous" uses two architectures with similar generative capabilities.

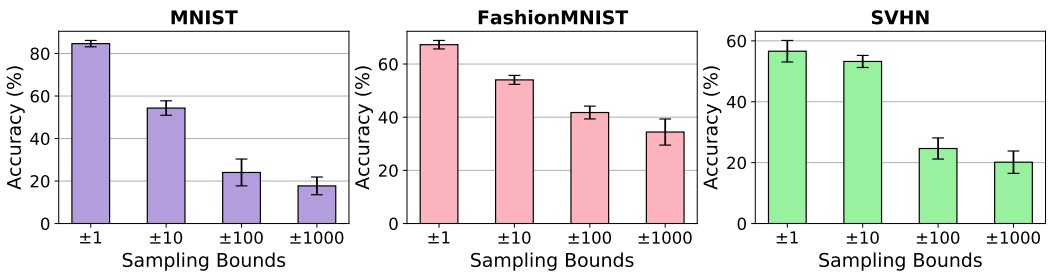

Figure 5: Results for our proposed security-promoting extension at high heterogeneity ($\alpha = 0.05$). We show the accuracy for a classifier trained on samples from the intercepted client decoders and using client label distributions when the center of the normal prior is unknown. The sampling bounds represent the parameters of the uniform distribution used for latent vector sampling, i.e., $\pm 100$ represents the distribution $\mathcal{U}(-100, 100)$. The prior is a multivariate standard normal $\mathcal{N}(\mathbf{0}, \mathbf{I})$.

### 4.3 EXTENSIONS

**Heterogeneous Local Models.** To simulate clients with diverse computational resources, we train both FEDCVAE-ENS and FEDCVAE-KD using two local CVAE architectures: the first as described in Section 4.1 and the second with one convolutional/deconvolutional block removed for the encoder/decoder, respectively. The server decoder matches that which is described in Section 4.1 but can be chosen arbitrarily. Because the two architectures demonstrate similar generative capabilities, final server classifier accuracy is very similar when comparing homogeneous against heterogeneous local architectures (Figure 4).[4] Notably, our KD procedure for FEDCVAE-KD still performs well using heterogeneous models, indicating diverse architectures can organize latent space similarly enough to successfully translate this knowledge to a single decoder.

**Promoting Security.** We verify the effectiveness of our proposed distribution shift extension for securing the uploaded information. Suppose an eavesdropping attacker is able to intercept the label distributions, decoder weights, and decoder architectures from all clients during upload. Without knowledge of the shared center of the multivariate normal prior $\boldsymbol{\mu}$, we show that training a performant classifier is infeasible because it is difficult to extract high-quality samples from the client decoders. In particular, even when the attacker samples latent vectors $\mathbf{z}$ from a broad region which overlaps with the high-density region of the prior (i.e., a uniform distribution centered on $\boldsymbol{\mu}$), the accuracy of the classifier trained on the resulting samples degrades sharply as the sampling region

---

[4]We choose not to baseline against FEDONESHOT, which supports heterogeneous local models, because our notion of "heterogeneous model" is different; FEDONESHOT supports heterogeneous local classifiers while our two methods defer the choice of classifier to after FL is complete and instead support heterogeneous local CVAEs.

grows (Figure 5). Even a good guess of $\mathcal{U}(-10, 10)$ for a normal prior of $\mathcal{N}(\mathbf{0}, \mathbf{I})$ results in a $3-30\%$ point decrease in accuracy depending on the dataset. A more realistic guess of $\mathcal{U}(-1000, 1000)$ results in a $33-67\%$ point decrease. Therefore, this simple FEDCVAE-ENS and FEDCVAE-KD extension reduces eavesdropping attackers' capacity to extract high-quality samples from uploaded decoders or train a performant downstream model, reducing communication risks.

## 5    RELATED WORK

**FL Under High Statistical Heterogeneity.** FL was originally proposed by McMahan et al. (2017) as a paradigm for decentralized distributed learning. Statistical heterogeneity quickly emerged as a core issue within FL, with many studies focusing on maintaining high performance under very non-IID data. Many approaches have experimented with augmenting FEDAVG by adding proximal terms to the local objective as an attempt to restrain local updates (Li et al., 2020c; Karimireddy et al., 2020; Wang et al., 2020; Acar et al., 2021; Li et al., 2021a). Other works have used KD to circumvent issues associated with parameter averaging under non-IID data, focusing on leveraging auxiliary data (Lin et al., 2020; Sattler et al., 2021) or a generative model (Zhu et al., 2021; Zhang et al., 2022) to compactly capture client ensemble learning. Recent approaches focus on improving client selection strategy (Tang et al., 2022), reducing catastrophic forgetting to balance global knowledge against local learning (Huang et al., 2022), or improving the generality of client models (Mendieta et al., 2022). However, all of these methods are designed for standard FL and rely heavily on local regularization through substantial iterative communication, which is not feasible in the one-shot FL setting where communication is limited to a single round.

**One-Shot FL.** Methods in one-shot FL have demonstrated strong performance under substantial communication constraint. Guha et al. (2019) originally proposed one-shot FL and introduced two methods, one based on heuristic selection methods for client inclusion in the final ensemble and another that used KD with an auxiliary dataset for ensemble aggregation. Li et al. (2021b) extended the use of KD with a hierarchical KD procedure with wide applicability to a variety of local model types. Although these methods obtain high accuracy, they rely on an auxiliary public dataset to support KD, which is inapplicable in data-free FL. Secure dataset transfer has been applied in several studies: Zhou et al. (2020) achieved this through dataset distillation, which requires a shared model architecture, and Shin et al. (2020) provided limited experiments using XOR-based data augmentation techniques. Zhang et al. (2021) proposed a data-free KD procedure based on a generator network trained using the ensemble of client classifiers, showing promising performance even under heterogeneous local models. However, existing approaches in one-shot FL either do not experiment with high statistical heterogeneity or degrade even under moderate levels of heterogeneity, unlike our proposed methods.

**VAEs in FL.** A few studies have experimented with using VAEs in FL. Kasturi et al. (2022) proposed a distributed learning framework based on VAEs, but required an auxiliary pre-trained classifier to generate sample labels and did not specify how to obtain this classifier. Wen et al. (2020) and Gu & Yang (2021) used CVAEs to protect against malicious clients, but only included limited experimental results with respect to statistical heterogeneity, relied on multiple communication rounds, and, in the case of Wen et al. (2020), did not use KD for server-side aggregation. We are the first to apply CVAEs to one-shot FL with a focus on high statistical heterogeneity.

## 6    CONCLUSION

In this paper, we proposed FEDCVAE-ENS and FEDCVAE-KD, data-free one-shot FL methods that reframe the local learning task using CVAEs. Both methods performed well given high statistical heterogeneity, demonstrated consistent performance with increasing numbers of clients, allow for model heterogeneity across clients, and can be extended to promote security. Extensive experimental results showed that FEDCVAE-ENS and FEDCVAE-KD substantially extended the state-of-the-art in one-shot FL under very high statistical heterogeneity, making way for more nuanced development in this difficult environment for distributed learning.

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

## A    ADDITIONAL DESCRIPTION OF METHODS

**FEDCVAE-ENS Description.** FEDCVAE-ENS follows the same procedure as FEDCVAE-KD, but alternatively defines $\mathcal{D}_{\text{Ens}}$ as the combination of client subsets $\mathcal{D}_{\text{Ens}}^k := \{(\hat{\mathbf{x}}_i^k, \mathbf{y}_i^k)\}_{i=1}^{\lfloor n_C/m \rfloor}$ and uses $\mathcal{D}_{\text{Ens}}$ to directly train the server classifier. Figure 6 visualizes the full pipeline for FEDCVAE-ENS and Algorithm 2 details the full procedure. Algorithm 3 shows the local client training procedure, which is the same for FEDCVAE-ENS and FEDCVAE-KD. Note that while we represent parameter optimization as using stochastic gradient descent in all algorithms, any optimizer can be used; for our experiments, we uniformly use Adam (Kingma & Ba, 2014).

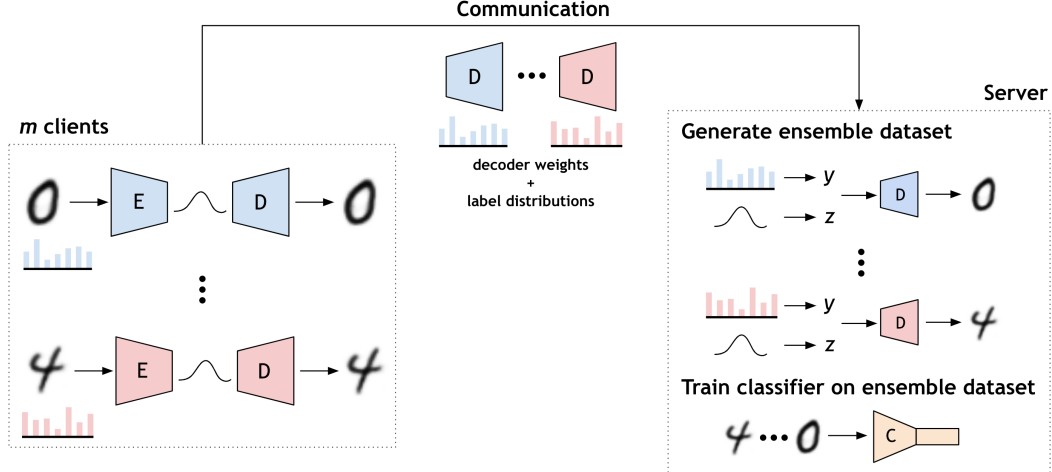

Figure 6: The full pipeline for one of our proposed methods, FEDCVAE-ENS. Here, E, D, and C represent "encoder," "decoder," and "classifier" models, respectively. First, clients train CVAEs on their private local datasets. Then, the server uses the ensemble of uploaded client decoders and corresponding local label distributions to generate a labeled dataset of synthetic samples to train a classifier.

---

**Algorithm 2 - FEDCVAE-ENS in the one-shot FL setting.** $T_L$ represents the number of local training epochs. The server classifier parameters are $\mathbf{w}_C^S$, with training epochs $T_C$, number of training samples $n_C$, classification loss $\ell_C(\cdot)$, and learning rate $\eta_C$. $\mathcal{C}$ is the set of clients.

---
1: **procedure** SERVER
2:     **for** each client $k \in \mathcal{C}$ **in parallel do**
3:         $\boldsymbol{\theta}^k,\ \hat{p}^k(\mathbf{y}) \leftarrow$ **ClientLocalUpdate**$(k, T_L)$
4:     Generate samples from each client $\mathcal{D}_{\text{Ens}}^k := \{(\hat{\mathbf{x}}_i^k, \mathbf{y}_i^k)\}_{i=1}^{\lfloor n_C/m \rfloor}$ using client decoder $D_{\boldsymbol{\theta}^k}(\cdot)$
            and label distribution $\hat{p}^k(\mathbf{y})$
5:     Combine client subsets into an IID labeled dataset $\mathcal{D}_{\text{Ens}} := \mathcal{D}_{\text{Ens}}^1 \cup \mathcal{D}_{\text{Ens}}^2 \cup ... \cup \mathcal{D}_{\text{Ens}}^m$
6:     **for** classifier epoch $i = 1$ to $T_C$ **do**
7:         **for** mini-batch $b \subset \mathcal{D}_{\text{Ens}}$ **do**
8:             $\mathbf{w}_C^S \leftarrow \mathbf{w}_C^S - \eta_C \cdot \nabla_{\mathbf{w}_C^S} \ell_C(\mathbf{w}_C^S; b)$
---

## B    DETAILS ON THE PRIVACY- AND SECURITY-PROMOTING EXTENSIONS

**Privacy.** A private pipeline is one that does not leak private client data to other participating clients or the server. van den Burg & Williams (2021) define a probabilistic generative model's propensity to reproduce samples observed in the training data as memorization, and prove that ensuring a particular level of differential privacy (DP) can bound memorization in probabilistic generative models (including CVAEs). Thus, beyond the normal privacy guarantees attributable to DP, employing FL DP-estimation techniques (i.e., Geyer et al. (2017)) also ensures low memorization across

---

**Algorithm 3** - **Local training procedure.** $\mathcal{D}^k$ represents the client's local dataset. The local learning rate is $\eta$ and the local loss function is $\ell(\cdot)$

---

1: **procedure** CLIENTLOCALUPDATE($k, T_L$)
2:      Initialize local CVAE parameters $\mathbf{w}^k := [\boldsymbol{\phi}^k, \boldsymbol{\theta}^k]$
3:      **for** local epoch in $i = 1$ to $T_L$ **do**
4:          **for** mini-batch $b \subset \mathcal{D}^k$ **do**
5:              $\mathbf{w}^k \leftarrow \mathbf{w}^k - \eta \cdot \nabla_{\mathbf{w}^k} \ell(\mathbf{w}^k; b)$          ▷ Optimize based on Equation 1
     **return** Decoder parameters $\boldsymbol{\theta}^k$ and local label distribution $\hat{p}^k(\mathbf{y})$ to the server

---

the FEDCVAE-ENS and FEDCVAE-KD pipelines, further enhancing the privacy of our proposed methods. We leave further exploration of privacy-preserving extensions to future work.

**Quality of Trained Decoder Samples.** To generate high-quality samples from a trained CVAE, it is typical to sample latent vectors $\mathbf{z}_i$ either directly from the prior (usually a multivariate standard normal, i.e., $\mathbf{z}_i \sim \mathcal{N}(\mathbf{0}, \mathbf{I})$) or from some other distribution with tight bounds around the prior distribution's mean (e.g., a truncated standard normal or a uniform distribution). During training, the CVAE will largely observe latent vectors in the highest density region of the prior distribution; for a standard normal distribution, this is near the center $\boldsymbol{\mu} = \mathbf{0}$. Latent vectors distant from the center will not generate high-quality samples when used with the trained decoder. As a demonstration, we train a centralized CVAE and sample both close to the center of the prior (i.e., $\mathbf{z}_i \sim \mathcal{U}(-1, 1)$) and distant from the center of the prior (i.e., $\mathbf{z}_i \sim \mathcal{U}(5, 20)$). The resulting image samples are shown in Figure 7.

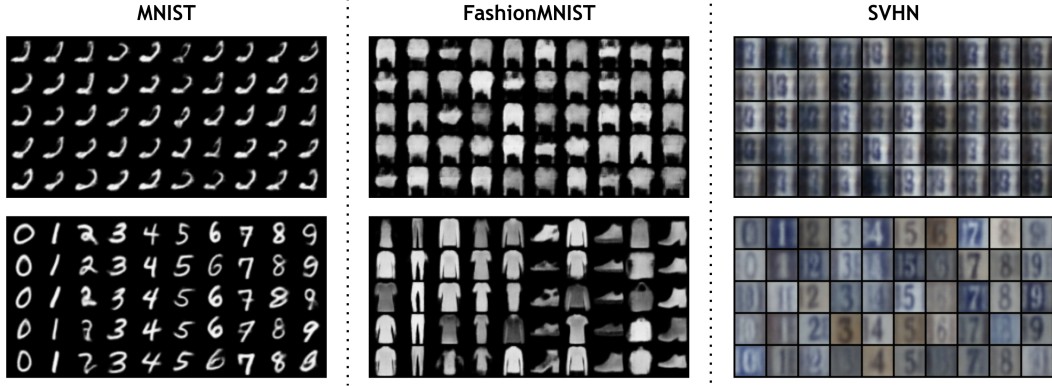

Figure 7: CVAE samples generated using latent vectors distant from the center of the multivariate normal prior (top row) and close to the center (bottom row).

## C    ADDITIONAL EXPERIMENTAL DETAILS AND RESULTS

**Benchmark Datasets.** MNIST and FashionMNIST contain $28 \times 28$ grayscale images of handwritten digits and clothing/accessories, respectively, with $60,000$ train samples and $10,000$ test samples. SVHN contains $32 \times 32$ RGB image crops of street-view house numbers, with $73,257$ train samples and $26,032$ test samples.

**Hyperparameter Settings.** Tables 3 and 4 contain fixed and variable hyperparameters, respectively. While the number of local epochs ($T_L$) may seem low for FEDAVG, we observed substantially reduced accuracy at higher numbers of local epochs, which is consistent with Lin et al. (2020) and references therein.

**Stability With Respect to Dataset Partition.** To complement the results in Table 2, we test the stability of each model across varying dataset partitions, controlled by a random seed (Table 5). When $\alpha$ is very low, as in our study, the dataset client splits generated by sampling from the Dirichlet distribution are diverse. This is exacerbated when more clients are used (higher $m$), potentially explaining some of the unstable results in Table 2. While FEDAVG and FEDONESHOT are very

Table 3: The default hyperparameter settings, which are used in experiments unless otherwise mentioned. These values are held consistent across datasets.

|  | Hyperparameter | Value |
|---|---|---|
| Shared | Local learning rate ($\eta$) | 0.001 |
|  | Classifier optimizer | Adam |
|  | Batch size (all) | 32 |
| FEDCVAE-ENS & FEDCVAE-KD | Server decoder optimizer | Adam |
|  | CVAE optimizer | Adam |
|  | Server classifier train samples ($n_C$) | 5000 |
|  | Server decoder train samples ($n_D$) | 5000 |
|  | Server classifier learning rate ($\eta_C$) | 0.001 |
|  | Server decoder learning rate ($\eta_D$) | 0.01 |

Table 4: Dataset-specific hyperparameter settings, where applicable. "Truncated normal width" denotes the truncation bounds for the truncated standard normal distribution used for sampling from client decoders. The truncated normal width values are in terms of number of standard deviations.

|  | Hyperparameter | MNIST | FashionMNIST | SVHN |
|---|---|---|---|---|
| FEDAVG | Local epochs ($T_L$) | 10 | 10 | 5 |
| FEDONESHOT | Local epochs ($T_L$) | 15 | 15 | 10 |
| FEDCVAE-ENS | Local epochs ($T_L$) | 15 | 25 | 50 |
|  | CVAE latent dimension ($d$) | 10 | 10 | 10 |
|  | Server classifier epochs ($T_C$) | 10 | 5 | 5 |
|  | Truncated normal width | $\pm 3$ | $\pm 3$ | $\pm 3$ |
| FEDCVAE-KD | Local epochs ($T_L$) | 15 | 25 | 50 |
|  | CVAE latent dimension ($d$) | 10 | 100 | 10 |
|  | Server classifier epochs ($T_C$) | 10 | 5 | 5 |
|  | Server decoder epochs ($T_D$) | 7 | 10 | 80 |
|  | Truncated normal width | $\pm 1$ | $\pm 2$ | $\pm 3$ |

sensitive to dataset split (high standard deviation), the results in Table 5 for both FEDCVAE-ENS and FEDCVAE-KD are not only consistent with Table 2, but also exhibit reasonable stability (low standard deviation).

Table 5: Performance of four one-shot FL methods over three datasets and across four numbers of clients $m$. Results show the average test accuracy across 5 random dataset partitions $\pm$ one standard deviation. Parameter initialization remains constant. Best results for each dataset and each level of $m$ are in purple , with second best results in yellow .

|  | # of Clients | FEDAVG | FEDONESHOT | FEDCVAE-KD (ours) | FEDCVAE-ENS (ours) |
|---|---|---|---|---|---|
| MNIST | $m = 5$ | $50.58 \pm 8.12$ | $38.81 \pm 6.49$ | $80.84 \pm 1.03$ | $93.97 \pm 1.11$ |
|  | $m = 10$ | $52.67 \pm 12.67$ | $34.90 \pm 10.79$ | $82.67 \pm 0.85$ | $94.01 \pm 1.54$ |
|  | $m = 20$ | $38.29 \pm 4.71$ | $38.85 \pm 9.04$ | $82.08 \pm 0.89$ | $93.29 \pm 2.16$ |
|  | $m = 50$ | $30.24 \pm 7.24$ | $45.65 \pm 14.79$ | $79.49 \pm 1.60$ | $89.68 \pm 2.21$ |
| FashionMNIST | $m = 5$ | $44.87 \pm 7.02$ | $36.28 \pm 8.04$ | $65.60 \pm 3.06$ | $71.80 \pm 3.59$ |
|  | $m = 10$ | $37.54 \pm 13.00$ | $34.26 \pm 6.00$ | $69.43 \pm 2.26$ | $75.42 \pm 2.47$ |
|  | $m = 20$ | $34.34 \pm 12.70$ | $33.57 \pm 8.59$ | $69.55 \pm 1.47$ | $76.66 \pm 1.79$ |
|  | $m = 50$ | $28.89 \pm 8.43$ | $36.49 \pm 7.65$ | $68.91 \pm 0.30$ | $74.70 \pm 2.09$ |
| SVHN | $m = 5$ | $30.27 \pm 7.88$ | $35.01 \pm 6.02$ | $57.43 \pm 2.20$ | $66.01 \pm 1.30$ |
|  | $m = 10$ | $28.92 \pm 10.46$ | $33.80 \pm 3.81$ | $55.29 \pm 3.45$ | $65.34 \pm 1.32$ |
|  | $m = 20$ | $20.63 \pm 2.16$ | $29.92 \pm 12.12$ | $49.18 \pm 2.84$ | $65.82 \pm 1.16$ |
|  | $m = 50$ | $19.59 \pm 0.01$ | $30.22 \pm 2.83$ | $38.83 \pm 2.97$ | $64.91 \pm 1.85$ |

**Comparing KD With Parameter Averaging.** FEDAVG (McMahan et al., 2017) introduced the notion of parameter averaging to FL. While parameter averaging may seem like a reasonable approach for client decoder aggregation in FEDCVAE-KD, it generates qualitatively poor samples (Figure 8) and fails to train a high-accuracy classifier (Table 6). The KD approach we propose to aggregate client decoders generates substantially better samples (Figure 8) while also obtaining more than $2\times$ classifier accuracy on MNIST, more than $3.25\times$ accuracy on FashionMNIST, and nearly $5\times$ accuracy for SVHN at $\alpha = 0.001$ (Table 6).

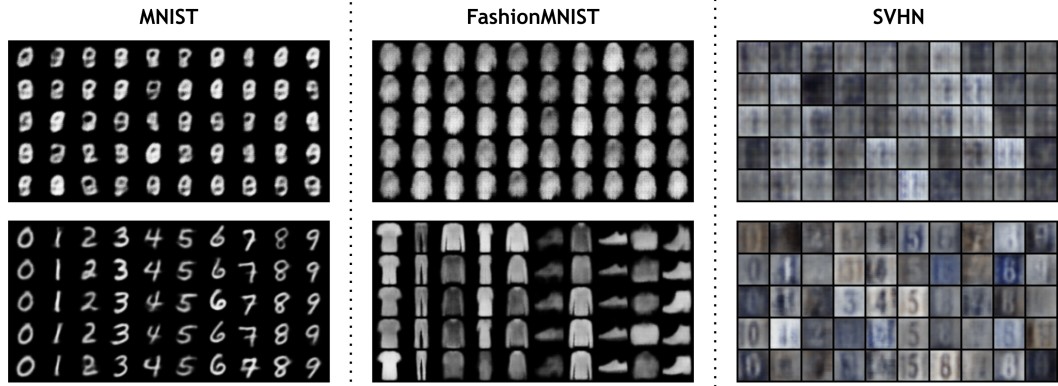

Figure 8: Samples from the aggregated server decoder obtained through parameter averaging (top row) versus knowledge distillation (bottom row) at high statistical heterogeneity ($\alpha = 0.01$).

Table 6: Comparing the knowledge distillation aggregation methods of FEDCVAE-KD to simple parameter averaging. Best results for each dataset and each level of $\alpha$ are in purple.

|  | Heterogeneity | Parameter Averaging | Knowledge Distillation |
|---|---|---|---|
| MNIST | $\alpha = 0.001$ | $38.59 \pm 3.29$ | $82.24 \pm 1.09$ |
|  | $\alpha = 0.01$ | $39.50 \pm 6.76$ | $82.01 \pm 1.61$ |
|  | $\alpha = 0.05$ | $50.20 \pm 4.63$ | $79.57 \pm 1.17$ |
| FashionMNIST | $\alpha = 0.001$ | $20.88 \pm 4.78$ | $69.53 \pm 1.70$ |
|  | $\alpha = 0.01$ | $27.66 \pm 7.08$ | $69.97 \pm 1.63$ |
|  | $\alpha = 0.05$ | $39.35 \pm 6.27$ | $67.24 \pm 1.96$ |
| SVHN | $\alpha = 0.001$ | $11.26 \pm 0.77$ | $55.48 \pm 2.12$ |
|  | $\alpha = 0.01$ | $15.22 \pm 4.29$ | $55.97 \pm 0.38$ |
|  | $\alpha = 0.05$ | $10.61 \pm 2.23$ | $54.24 \pm 2.08$ |

**Performance With Less Data Per Client.** To gauge our proposed methods' performance relative to the size of each client's local dataset, we vary the percent of the benchmark training data distributed to clients (Table 7). While accuracy for both FEDCVAE-ENS and FEDCVAE-KD degrade with less data per client, they both consistently perform better than FEDAVG and FEDONESHOT across all tested percent subsets of the training data.

**Adding Noise to Uploaded Label Distributions.** Uploading client label distributions $\hat{p}^k(\mathbf{y})$ may generate additional privacy concerns. One potential solution is to mask the precise label counts for each client by adding noise before upload as in Zhang et al. (2022); to achieve this, we draw noise from a normal distribution $\epsilon_c \sim \mathcal{N}(0, \gamma \cdot n_k)$ such that the "strength" (variance) of the noise applied to class $c$ is in proportion $\gamma$ to the total number of training samples for client $k$. Noise is applied to the training sample count for each class for a given client before uploading this information to the server. We visualize the effect of noise on client label distributions for several levels of $\gamma$ in Figure 9. As $\gamma \to \infty$, information from the uploaded label distribution disappears; when $\gamma = 0$, the exact local label distributions are communicated. When adding a modest amount of noise to the client label distributions (i.e., $\gamma \leq 0.1$), accuracy for both FEDCVAE-ENS and FEDCVAE-KD is barely affected compared to upload with no noise; see Table 8 for $\gamma > 0$ and refer to the $\alpha = 0.01$

Table 7: Performance of four data-free one-shot FL methods over three datasets and across multiple percent subsets of each dataset. Best results for each dataset and percent subset are in purple, with second best results in yellow.

| | % Subset | FEDAVG | FEDONESHOT | FEDCVAE-KD (ours) | FEDCVAE-ENS (ours) |
|---|---|---|---|---|---|
| MNIST | 5% | $40.65 \pm 2.65$ | $34.24 \pm 1.51$ | $74.98 \pm 4.82$ | $76.47 \pm 2.58$ |
| | 10% | $38.55 \pm 4.43$ | $36.86 \pm 1.14$ | $81.24 \pm 1.45$ | $89.95 \pm 1.27$ |
| | 25% | $56.17 \pm 3.82$ | $47.50 \pm 0.72$ | $80.00 \pm 1.52$ | $92.41 \pm 0.79$ |
| | 50% | $58.29 \pm 2.83$ | $41.15 \pm 0.70$ | $82.01 \pm 1.61$ | $93.83 \pm 1.53$ |
| FashionMNIST | 5% | $14.11 \pm 2.33$ | $26.96 \pm 0.74$ | $66.75 \pm 1.80$ | $69.78 \pm 1.26$ |
| | 10% | $26.53 \pm 6.28$ | $27.26 \pm 1.69$ | $69.07 \pm 1.53$ | $72.10 \pm 1.66$ |
| | 25% | $28.29 \pm 1.69$ | $29.97 \pm 2.53$ | $71.24 \pm 1.48$ | $75.31 \pm 1.42$ |
| | 50% | $45.85 \pm 2.95$ | $37.63 \pm 0.53$ | $69.97 \pm 1.63$ | $76.62 \pm 1.61$ |
| SVHN | 5% | $17.19 \pm 5.38$ | $11.08 \pm 0.38$ | $41.74 \pm 3.67$ | $46.06 \pm 1.29$ |
| | 10% | $22.97 \pm 4.37$ | $16.16 \pm 0.73$ | $47.65 \pm 2.88$ | $54.36 \pm 1.76$ |
| | 25% | $28.40 \pm 3.15$ | $29.10 \pm 0.26$ | $52.15 \pm 1.23$ | $59.96 \pm 2.68$ |
| | 50% | $20.38 \pm 1.03$ | $32.39 \pm 2.85$ | $57.56 \pm 1.09$ | $64.82 \pm 1.27$ |
| | 100% | $25.12 \pm 2.07$ | $31.91 \pm 1.26$ | $55.97 \pm 0.38$ | $65.50 \pm 0.28$ |

Table 8: Performance of our proposed methods with noise added to the uploaded client label distributions. Results show the average test accuracy across 5 seeds for the random noise $\pm$ one standard deviation.

| | Noise Proportion | FEDCVAE-KD | FEDCVAE-ENS |
|---|---|---|---|
| MNIST | $\gamma = 0.01$ | $81.10 \pm 0.69$ | $93.69 \pm 0.82$ |
| | $\gamma = 0.05$ | $80.41 \pm 1.81$ | $92.65 \pm 0.77$ |
| | $\gamma = 0.1$ | $80.56 \pm 0.36$ | $92.65 \pm 1.33$ |
| FashionMNIST | $\gamma = 0.01$ | $69.43 \pm 1.61$ | $76.44 \pm 1.11$ |
| | $\gamma = 0.05$ | $68.57 \pm 1.74$ | $74.44 \pm 2.14$ |
| | $\gamma = 0.1$ | $69.07 \pm 1.97$ | $72.88 \pm 3.03$ |
| SVHN | $\gamma = 0.01$ | $57.18 \pm 2.24$ | $64.03 \pm 1.78$ |
| | $\gamma = 0.05$ | $56.93 \pm 2.43$ | $62.68 \pm 2.64$ |
| | $\gamma = 0.1$ | $55.85 \pm 2.29$ | $61.47 \pm 1.83$ |

in Table 1 for $\gamma = 0$. The notion of uploading and harnessing client label distributions in FL is new (Zhu et al., 2021; Zhang et al., 2022) and quantifying the privacy risks that label distributions might induce is an open problem which could benefit from focused development.

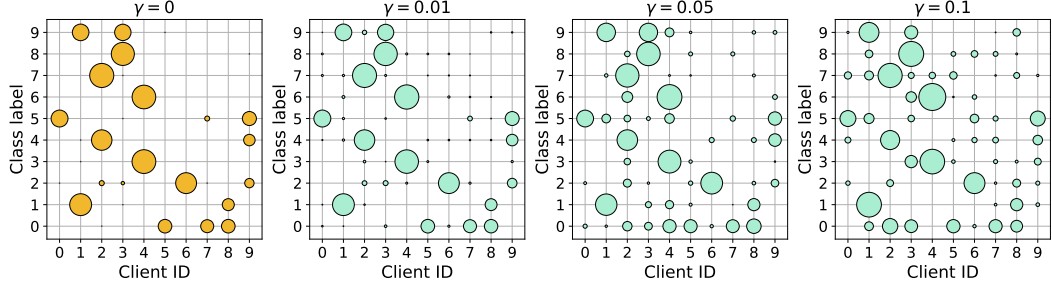

Figure 9: The effect of the noise proportion $\gamma$ on an example data partition at $\alpha = 0.01$, $m = 10$ clients, and on MNIST. The original partition is in yellow ($\gamma = 0$). The size of each dot is proportional to the number of samples.

