# OpenReview forum: "Data-Free One-Shot Federated Learning Under Very High Statistical Heterogeneity"
_ICLR.cc/2023/Conference — ICLR 2023 poster_

### Official Review · Reviewer_feTx · 2022-10-21

**Confidence:** 4
**Correctness:** 4
**Technical Novelty And Significance:** 2
**Empirical Novelty And Significance:** 3
**Recommendation:** 6

**Clarity, Quality, Novelty And Reproducibility:**

This work is written with clarity and good quality. The originality of this work is good.

**Strength And Weaknesses:**

Strength:

-- The proposed solutions seem effective under the one-shot FL setting. The efficacy and efficiency are both demonstrated in the extensive experiments.
-- The writing is clear and easy to follow. The tables and figures are well-organized.

Weaknesses:

-- Constructing a generative model to generate samples for FL model training is not new. There are a few works utilizing a similar strategy in FL, for example:
Data-Free Knowledge Distillation for Heterogeneous Federated Learning, ICML 2021
Fine-tuning global model via data-free knowledge distillation for non-iid federated learning, CVPR 2022
Please make a comparison with those works

-- Some details are not clear enough for me. For example, in Eq 2, what is the formulation for g(.)?  Is this dissimilarity measured in the data-space or feature space?

**Summary Of The Paper:**

This paper targets to one-shot federated learning under high statistical heterogeneity. The authors designed two different solutions: FEDCVAE-ENS and FEDCVAE-KD, both of which construct a VAE on the server to generate samples. The generated samples are utilised to train a classifier on the server. The difference between FEDCVAE-ENS and FEDCVAE-KD is that FEDCVAE_KD create a single server VAE decoder while FEDCVAE-ENS stores a set of VAEs that is trained on local clients.

**Summary Of The Review:**

This work focuses on one-shot FL with high data heterogeneity. Two solutions are proposed in this work, which has been tested under extensive experiments. The writing is clear and well-organized. There might be some minor issues in this draft. I look forward to the feedback from the authors in the rebuttal.

---

> ### Author Response · Authors · 2022-11-16
> **Response to Reviewer feTx**
>
> **Q 4.1: Constructing a generative model to generate samples for FL model training is not new. There are a few works utilizing a similar strategy in FL, for example: "Data-Free Knowledge Distillation for Heterogeneous Federated Learning" (ICML 2021) and "Fine-tuning global model via data-free knowledge distillation for non-iid federated learning" (CVPR 2022). Please make a comparison with those works.**
>
> **A 4.1:** We would like to point out the significant differences between our method and existing studies. We discussed [Zhu 2021] in the Related Works (Section 5). The approach is fundamentally different from our proposed methods, since the resulting generator in [Zhu 2021] is used to extract global information from classifiers to regularize local training, and generates samples at a reduced intermediate feature level rather than in full data space. Additionally, we added a brief comparison to [Zhang 2022] to the Related Works (Section 5). Again, the generative model in [Zhang 2022] does not attempt to directly capture the data distribution and is instead used to facilitate knowledge distillation for client classifier aggregation. Our method diverges substantially from both [Zhu 2021] and [Zhang 2022] because of the generative local learning task and the deferral of classifier training until after the full federated procedure is complete, offering flexibility in the downstream task.
>
> **References:**
>
> [Zhu 2021] Zhu, Z., Hong, J. \& Zhou, J. Data-Free Knowledge Distillation for Heterogeneous Federated Learning. In Proceedings of the 38th International Conference on Machine Learning (PMLR, 2021).
>
> [Zhang 2022] Zhang, L., Shen, L., Ding, L., Tao, D. \& Duan, L.-Y. Fine-Tuning Global Model via Data-Free Knowledge Distillation for Non-IID Federated Learning. In Proceedings of the IEEE/CVF Conference on Computer Vision and Pattern Recognition (CVPR, 2022).
>
> **Q 4.2: Some details are not clear enough for me. For example, in Eq 2, what is the formulation for $g(\cdot)$? Is this dissimilarity measured in the data-space or feature space?**
>
> **A 4.2:** The dissimilarity function $g(\cdot)$ of Equation 2 is measured in data space; as noted on Page 5, $g(\cdot)$ measures the dissimilarity between the synthetic data sample generated by the server decoder $D_{\boldsymbol\theta^S}(\mathbf{z}^k; \mathbf{y}^k)$ and the client decoder data sample $\hat{\mathbf{x}}^k$ for the same latent vector $\mathbf{z}^k$ and conditioning class $\mathbf{y}^k$. We added this clarification to Section 3.2. Please let us know if there are any other details that need clarification.

---

> ### Author Response · Authors · 2022-11-29
> **Thank you for your feedback and reminder to view our rebuttal**
>
> Reviewer feTx,
>
> Thank you for your comments and constructive criticism on the initial version of the manuscript. We have revised the manuscript according to your comments, in particular adding a discussion of FL methods that use generative models and clarifying the formulation for the dissimilarity function $g(\cdot).$ Please let us know if you have any additional concerns so that we can address them as soon as possible. Thank you!

---

### Official Review · Reviewer_TDXe · 2022-10-24

**Confidence:** 4
**Correctness:** 3
**Technical Novelty And Significance:** 2
**Empirical Novelty And Significance:** 2
**Recommendation:** 6

**Clarity, Quality, Novelty And Reproducibility:**

The paper presents a relatively simple idea and is mostly clearly written. As far as I am aware, this method is novel in the context of FL. The authors also provide a reasonable amount of information, which should make FedCVAE reproducible. The authors seem to mostly tackle the cross-silo setting, where each client has enough data. One thing I noticed is that the authors mention that the conditional VAE (CVAE) objective at eq. 1 bounds the marginal likelihood $\log p(x)$; this is not correct, as the specific objective at eq. 1 bounds the conditional marginal likelihood $\log p(x|y)$.

Besides that, there are a couple of things that are, in my opinion, not properly motivated. More specifically, the authors propose to sample from the generative model and train a classifier at the server. Why is this necessary and why you do not simply use the generative classifier  $p(y|x) = \frac{\int p(x| z, y)p(z)dz p(y)}{p(x)}$? To me it seems that the target for the auxiliary classifier is nothing else, but to approximate this generative classifier. The loss for the auxiliary classifier (in the limit where you constantly sample from the generative model) can be obtained by minimising the average KL-divergence between the true CVAE generative posterior $p(y|x)$ and the classifier predictive $q(y|x)$. More specifically: $$E_p(x)[KL(p(y|x) || q(y|x))] = - E_p(y,x) [\log q(y|x)] + const = -E_p(y,z,x)[\log q(y|x)]  + const = -E_p(y)p(z)p(x|y,z)[\log q(y|x)] + const$$ In this sense, it seems that 1) it might not be necessary to train an auxiliary classifier and 2) the modelling choices made in the generative model itself are important for the performance of the actual downstream classifier. I believe a discussion about this is warranted to be added in the paper. Apart from that, when the authors want to sample datapoints from the model they actually use a different distribution over the latents than the one the model was actually trained; e.g., using a $z \sim U[-1, 1]$ instead of $z\sim N(0, I)$. They mention that it is beneficial to stay at a high density region of the prior, but how does the performance of the algorithm fare when you use the correct prior? Why not use e.g., $z \sim N(0, \sigma)$ with $\sigma < 1$ which is closer to the correct prior?

Another thing that could be better explained are the security and privacy aspects of this method, especially given the FL scenario. FedCVAE uploads the CVAE decoder along with $p(y)$ to the server, which is inherently less private than FedAvg. While it is true that with DP you can bound memorisation and get privacy, at the moment the authors do not use anything and the gains in performance when doing DP relatively to e.g., DP-FedAvg are not clear. In fact, given a sufficient powerful generative model, this method should be indistinguishable from sending the actual data. Furthermore, the arguments about security by shifting $\mu$, need to be more formal. Are there some guarantees that you can give, given a sufficient powerful adversary? For example, given $p(y)$, and $p(x|z, y)$, couldn’t the attacker just try to infer a global mean for $z$? For example, by visual inspection of the samples or an optimization procedure? And a small comment; $U[-10, 10]$ is not an exceptionally good guess for $N(0, I)$ (which is what the text mentions at the beginning of page 9) as 99% of the probability of $N(0,I)$ is in $[-3, 3]$, so 70% of the mass of the uniform proposal is on almost zero probability mass under $N(0, I)$.

Furthermore, this method requires building good generative models which is generally a harder task than building a good discriminative model. How does this work on more complex datasets, such as CIFAR10 and CIFAR100, where $p(x|y, z)$ might not be as sharp? Furthermore, while non-iid settings are nice to have, to me it seems that this method might suffer in less non-iid / iid settings (e.g, $\alpha=1.0$, $\alpha=\inf$) ; the generative distribution is more diverse and thus it would be harder to model accurately with a, hardware constrained, VAE model. This can also sort of be seen when comparing FedCVAE-KD to FedCVAE-Ens; the former has worse performance that the latter, given that the entire training distribution needs to be modelled by a single model. How does the performance of the method scale with higher values of $\alpha$? Besides that, it seems that the evaluation against the baselines is not entirely apples-to-apples; for example, FedOneShot does not need to reveal to the server the labels and the models trained on the local data, so it has fundamentally different security / privacy tradeoffs.


**Strength And Weaknesses:**

Strengths
- Relatively simple procedure that requires a single round of communication
- As far as I know, using VAEs in this specific way for FL is novel

Weaknesses
- Simple datasets for generative models; how about CIFAR10, CIFAR100 or even Shakespeare?
- Experimental evaluation against baselines is not entirely apples to apples
- Some statements and explanations are not entirely accurate (see below)

**Summary Of The Paper:**

This work proposes an alternative way to perform federated learning; instead of training a discriminative model, the authors propose to train a conditional (on the label) VAE model $p(x, z| y) = p(z)p(x|z, y)$, to convergence on each client and then communicate (once), provided that the clients agree on a $p(z)$, the decoder along with the marginal label distribution of the client, back at the server. The server can then use the marginal $p(y)$ on each client along with its decoder to generate training data $(x, y)$ in order to train centrally a discriminative model. Having access to all of the decoders, the server can also train a global decoder to “distill” all of the client specific ones to reduce storage requirements. The authors further propose the use of a randomly centred prior mean for $p(z)$ in order to improve the security of the system against external entities. Better performance in cross-silo settings and high non-iid settings is demonstrated.


**Summary Of The Review:**

While the concept of FedCVAE is straightforward, it is novel in the federated setting. Having said that, given my aforementioned comments, around the auxiliary classifier, security / privacy and experimental evaluation, I cannot recommend acceptance of this work.

---

> ### Author Response · Authors · 2022-11-16
> **Response to Reviewer TDXe (1/5)**
>
> **Q 3.1: The authors mention that the conditional VAE (CVAE) objective at eq. 1 bounds the marginal likelihood $\text{log} \\, p(x)$; this is not correct, as the specific objective at eq. 1 bounds the conditional marginal likelihood $\text{log} \\, p(x | y)$.**
>
> **A 3.1:** Thank you for your careful reading. We corrected the indicated statement in the revised manuscript to reflect that Equation 1 bounds the conditional marginal likelihood $\text{log} \\, p(\mathbf{x} | \mathbf{y})$.
>
> **Q 3.2: The authors propose to sample from the generative model and train a classifier at the server. Why is this necessary and why you do not simply use the generative classifier $p(y|x) = \frac{\int p(x | y, z)p(z) dz \cdot p(y)}{p(x)}$? To me it seems that the target for the auxiliary classifier is nothing else, but to approximate this generative classifier. The loss for the auxiliary classifier (in the limit where you constantly sample from the generative model) can be obtained by minimising the average KL-divergence between the true CVAE generative posterior $p(y | x)$ and the classifier predictive $q(y | x)$. In this sense, it seems that 1) it might not be necessary to train an auxiliary classifier and 2) the modelling choices made in the generative model itself are important for the performance of the actual downstream classifier. I believe a discussion about this is warranted to be added in the paper.**
>
> **A 3.2:** As you point out, our proposed approach using an auxiliary classifier trained in the server does serve as a numerical approximation of the generative classifier. However, it would follow then that the performance of the auxiliary classifier serves as a lower bound on the accuracy of the generative classifier, i.e., the empirical results we obtain would not degrade under the full generative classifier. The choice of classifier is not central to our proposed methodology, which defers the terminal task to the end of the FL procedure to be more widely applicable. In fact, our models could extend to tasks like image segmentation, image imputation, and object counting. We chose to pursue classification as the central task in our paper to compare to one-shot FL baseline methods (i.e., FedAvg and FedOneShot). Per your suggestion, we added a brief discussion of the potential to directly use the generative classifier in the manuscript (please see Footnote 2 on Page 5), if other researchers want to commit to classification as the downstream task.
>
> Your second observation--that modeling choices in the generative model affect the downstream classifier--holds in the case of the auxiliary classifier as well. The generative capabilities of the chosen CVAE directly affect the model's capacity to capture each client's local data distribution, which impacts the classifier's capacity to discriminate between classes because it is trained only on CVAE samples. In this way, the generative modeling choices directly affect the quality of the auxiliary classifier.

---

> > ### Author Response · Authors · 2022-11-16
> > **Response to Reviewer TDXe (2/5)**
> >
> > **Q 3.3: Apart from that, when the authors want to sample datapoints from the model they actually use a different distribution over the latents than the one the model was actually trained; e.g., using a $\mathbf{z} \sim \mathcal{U}(-1, 1)$ instead of $\mathbf{z} \sim \mathcal{N}(\mathbf{0}, I)$. They mention that it is beneficial to stay at a high density region of the prior, but how does the performance of the algorithm fare when you use the correct prior? Why not use e.g., $z \sim \mathcal{N}(0, \sigma)$ with $\sigma < 1$ which is closer to the correct prior?**
> >
> > **A 3.3:** As suggested by Reviewer mEpo, we have conducted additional experiments with a truncated standard normal distribution for sampling synthetic data points from decoders, which is very similar to your suggestion of sampling using $\mathcal{N}(\mathbf{0}, \sigma \mathbf{I})$ with $\sigma < 1$. A tight uniform distribution over the highest density region of the normal prior was originally used simply because of improved empirical results during initial experiments. We conducted additional experiments using a standard normal distribution truncated to several bounds (i.e., $\pm 1$, $\pm 2$, and $\pm 3$ standard deviations) for sampling latent vectors. Qualitatively, we observed more diverse samples from the decoders when using the truncated normal compared to a uniform distribution, likely because of the closer match with the actual prior. This was reflected in a modest to substantial increase in accuracy across many of the experimental results. We have updated the pertinent figures and tables accordingly with the improved results in our revised version.
> >
> > Table 1 (also shown below) in the manuscript shows the performance of all models across varying levels of statistical heterogeneity using a truncated normal distribution as the prior. We note that FedCVAE-Ens now obtains over 95\% accuracy at $\alpha = 0.001$ on MNIST, and the best baseline method, FedAvg, achieves just 45.12\% accuracy. The greatest improvements can be seen for the dataset SVHN, with roughly a 10\% improvement in accuracy across all levels of $\alpha$ for FedCVAE-Ens and 6-7\% improvement for FedCVAE-KD. The results across the number of clients (Table 2 and Table 5 in the manuscript), heterogeneous local models (Figure 4 in the manuscript), various aggregation methods for FedCVAE-KD (Table 6 in the manuscript), and different percentage subsets of the data (Table 7 in the manuscript) have also been updated to reflect the use of a truncated normal distribution for decoder sampling. For more details, please refer to the revised manuscript.
> >
> > **Table:** Results for all methods using a truncated standard normal distribution for sampling from decoders for both of our proposed methods.
> > | Dataset      | Heterogeneity  |      FedAvg      |    FedOneShot    |    FedCVAE-KD    |    FedCVAE-Ens   |
> > |--------------|----------------|:----------------:|:----------------:|:----------------:|:----------------:|
> > | MNIST        | $\alpha=0.001$ | $45.12 \pm 5.87$ | $11.90 \pm 0.40$ | $82.24 \pm 1.09$ | $95.37 \pm 0.52$ |
> > | MNIST        | $\alpha=0.01$  | $58.29 \pm 2.83$ | $41.15 \pm 0.70$ | $82.01 \pm 1.61$ | $93.83 \pm 1.53$ |
> > | MNIST        | $\alpha=0.05$  | $80.10 \pm 2.35$ | $82.95 \pm 0.49$ | $79.57 \pm 1.17$ | $91.86 \pm 0.75$ |
> > | FashionMNIST | $\alpha=0.001$ | $32.77 \pm 4.52$ | $10.00 \pm 0.00$ | $69.53 \pm 1.70$ | $76.04 \pm 0.93$ |
> > | FashionMNIST | $\alpha=0.01$  | $45.85 \pm 2.95$ | $37.63 \pm 0.53$ | $69.97 \pm 1.63$ | $76.62 \pm 1.61$ |
> > | FashionMNIST | $\alpha=0.05$  | $46.86 \pm 2.37$ | $64.53 \pm 1.73$ | $67.24 \pm 1.96$ | $71.95 \pm 2.17$ |
> > | SVHN         | $\alpha=0.001$ | $20.31 \pm 4.36$ | $15.94 \pm 0.00$ | $55.48 \pm 2.12$ | $65.52 \pm 0.66$ |
> > | SVHN         | $\alpha=0.01$  | $25.12 \pm 2.07$ | $31.91 \pm 1.26$ | $55.97 \pm 0.38$ | $65.50 \pm 2.28$ |
> > | SVHN         | $\alpha=0.05$  | $33.66 \pm 4.12$ | $49.60 \pm 1.59$ | $54.24 \pm 2.08$ | $66.61 \pm 2.09$ |

---

> > > ### Author Response · Authors · 2022-11-16
> > > **Response to Reviewer TDXe (3/5)**
> > >
> > > **Q 3.4: Another thing that could be better explained are the security and privacy aspects of this method, especially given the FL scenario. FedCVAE uploads the CVAE decoder along with $p(y)$ to the server, which is inherently less private than FedAvg. While it is true that with DP you can bound memorisation and get privacy, at the moment the authors do not use anything and the gains in performance when doing DP relatively to e.g., DP-FedAvg are not clear. In fact, given a sufficient powerful generative model, this method should be indistinguishable from sending the actual data.**
> > >
> > > **A 3.4:** We follow a few highly-cited works [Zhu 2021, Zhang 2022] which use uploaded label distribution as a crucial element of the proposed method without excessive concern over violations of client privacy and which baseline against FedAvg. To address your concerns, we conducted additional experiments to elicit the relationship between the server classifier accuracy and the amount of information included in uploaded label distributions $\hat{p}^k(\mathbf{y})$. We follow [Zhang 2022] and add noise to the uploaded label distributions. We draw noise from a normal distribution $\epsilon_c \sim \mathcal{N}(0, \gamma \cdot n_k)$ such that the ``strength'' (variance) of the noise applied to class $c$ is in proportion $\gamma$ to the total number of training samples for client $k$. Noise is applied to the training sample count for each class for a given client before uploading this information to the server. As $\gamma \to \infty$, information from the uploaded label distribution disappears. When $\gamma = 0$, the exact local label distributions are communicated. Figure 9 in the revised manuscript visually shows how adding noise affects the label distributions. We also test the case where no local label distribution information is uploaded to the server, i.e., we uniformly sample conditioning classes when extracting synthetic samples from client decoders.
> > >
> > > The following tables include new experimental results for adding varying amounts of noise to the uploaded label distributions (Table 8 in the manuscript). The first of the tables below shows that when adding a modest amount of noise to the client label distributions (i.e., $\gamma \le 0.1$), accuracy for both FedCVAE-Ens and FedCVAE-KD is barely affected compared to uploading with no noise. Both proposed methods with varying amounts of noise continue to beat the baselines (compare to results in Table 1 of the manuscript where $\alpha = 0.01$). In the second table below, we display results of not uploading client label distributions to the server. Instead of using client label distributions for sampling conditioning classes $\mathbf{y}$, we use a uniform distribution. As expected, performance suffers, because the samples will be incorrectly labeled for the classes that the client decoder observed less, misleading the server classifier. However, we reiterate that small noise perturbations to the label distributions could offer improved privacy, and methods from previous work also rely on uploading client label distributions to the server.
> > >
> > > **Table:** Results for adding noise to the uploaded client label distributions.
> > > | Dataset    | Noise Proportion |   FedCVAE-KD   |   FedCVAE-Ens  |
> > > |--------------|------------------|:----------------:|:----------------:|
> > > | MNIST        | $\gamma=0.01$    | $81.10 \pm 0.69$ | $93.69 \pm 0.82$ |
> > > | MNIST        | $\gamma=0.05$    | $80.41 \pm 1.81$ | $92.65 \pm 0.77$ |
> > > | MNIST        | $\gamma=0.1$     | $80.56 \pm 0.36$ | $92.65 \pm 1.33$ |
> > > | FashionMNIST | $\gamma=0.01$    | $69.43 \pm 1.61$ | $76.44 \pm 1.11$ |
> > > | FashionMNIST | $\gamma=0.05$    | $68.57 \pm 1.74$ | $74.44 \pm 2.14$ |
> > > | FashionMNIST | $\gamma=0.1$     | $69.07 \pm 1.97$ | $72.88 \pm 3.03$ |
> > > | SVHN         | $\gamma=0.01$    | $57.18 \pm 2.24$ | $64.03 \pm 1.78$ |
> > > | SVHN         | $\gamma=0.05$    | $56.93 \pm 2.43$ | $62.68 \pm 2.64$ |
> > > | SVHN         | $\gamma=0.1$     | $55.85 \pm 2.29$ | $61.47 \pm 1.83$ |
> > >
> > > **Table:** Results for our proposed methods when client label distributions are not uploaded, i.e., uniform sampling across classes.
> > > | Dataset      | FedCVAE-KD       | FedCVAE-Ens      |
> > > |--------------|------------------|------------------|
> > > | MNIST        | $55.51 \pm 3.26$ | $60.71 \pm 2.59$ |
> > > | FashionMNIST | $27.84 \pm 4.85$ | $41.66 \pm 5.15$ |
> > > | SVHN         | $11.00 \pm 2.68$ | $18.09 \pm 6.27$ |
> > >
> > > As for the use of differential privacy (DP) to maintain client-level privacy as in [Geyer 2018], we do not believe that integrating DP would necessarily be straightforward in the context of our proposed methods because of the fundamentally different local learning task as compared to most FL methods (i.e., generative vs. discriminative). We leave the comparison to DP-FedAvg to future work. We do note that obtaining a generative model with sufficient power to *perfectly* capture the local data distributions is exceedingly unlikely in all but the easiest vision tasks.

---

> > > > ### Author Response · Authors · 2022-11-16
> > > > **Response to Reviewer TDXe (4/5)**
> > > >
> > > > **A 3.4 (continued)**
> > > >
> > > > **References:**
> > > >
> > > > [Zhu 2021] Zhu, Z., Hong, J. \& Zhou, J. Data-Free Knowledge Distillation for Heterogeneous Federated Learning. In Proceedings of the 38th International Conference on Machine Learning (PMLR, 2021).
> > > >
> > > > [Zhang 2022] Zhang, L., Shen, L., Ding, L., Tao, D. \& Duan, L.-Y. Fine-Tuning Global Model via Data-Free Knowledge Distillation for Non-IID Federated Learning. In Proceedings of the IEEE/CVF Conference on Computer Vision and Pattern Recognition (CVPR, 2022).
> > > >
> > > > [Geyer 2018] Geyer, R. C., Klein, T. \& Nabi, M. Differentially Private Federated Learning: A Client Level Perspective. Preprint at http://arxiv.org/abs/1712.07557 (2018).
> > > >
> > > > **Q 3.5: Furthermore, the arguments about security by shifting $\mu$, need to be more formal. Are there some guarantees that you can give, given a sufficient powerful adversary? For example, given $p(y)$, and $p(x | z, y)$, couldn't the attacker just try to infer a global mean for $z$? For example, by visual inspection of the samples or an optimization procedure? And a small comment; $\mathcal{U}(-10, 10)$ is not an exceptionally good guess for $\mathcal{N}(0, I)$ (which is what the text mentions at the beginning of page 9) as 99\% of the probability of $\mathcal{N}(0, I)$ is in $\mathcal{U}(-3, 3)$, so 70\% of the mass of the uniform proposal is on almost zero probability mass under $\mathcal{N}(0, I)$.**
> > > >
> > > > **A 3.5:** We believe the proposed security-preserving extension is a reasonable approach to *deter* would-be attackers. Inferring the global mean $\boldsymbol \mu$ via visual inspection would be exceptionally difficult given the vastness of real space; from the perspective of an attacker, this reduces to finding a tight region, which is related to the mean and variance of the normal prior, in all of $\mathbb{R}^d$. As we show in Appendix B, sampling too far outside of the high-density region of the prior distribution generates visually incoherent samples. Therefore, we believe the process of sampling from the client decoders and attempting to build $\boldsymbol \mu$ manually would be sufficiently time-intensive to deter an eavesdropping attacker. While perhaps there may be some optimization procedure to extract $\boldsymbol \mu$ more efficiently, we are not aware of a well-known procedure capable of this and do not feel extensive development in this area is within the scope of our paper. We could also combine this security-extension with the suggested DP-extension to further deter attackers. Furthermore, we revised the language for sampling from $\mathcal{U}(-10, 10)$ to reflect that, while a strong guess for $\boldsymbol \mu$ in all of $\mathbb{R}^d$, this is not an "exceptionally good guess" for the prior given its low overlap in terms of probability density (please see Section 4.3).
> > > >
> > > > **Q 3.6: This method requires building good generative models which is generally a harder task than building a good discriminative model. How does this work on more complex datasets, such as CIFAR10 and CIFAR100, where $p(x | y, z)$ might not be as sharp?**
> > > >
> > > > **A 3.6:** As requested, we have performed additional experiments on CIFAR-10; we used the default setting of $m = 10$ and distributed 100\% of the training set to clients at the highest levels of statistical heterogeneity (i.e., $\alpha = 0.01$ and $\alpha = 0.001$). In this setting, FedCVAE-Ens and FedCVAE-KD outperform the baselines.
> > > >
> > > > **Table:** Results for all methods on CIFAR-10 for multiple levels of $\alpha$.
> > > > |          | Heterogeneity  |      FedAvg      |    FedOneShot    |    FedCVAE-KD    |    FedCVAE-Ens   |
> > > > |----------|----------------|:----------------:|:----------------:|:----------------:|:----------------:|
> > > > | CIFAR-10 | $\alpha=0.001$ | $13.14 \pm 4.08$ | $16.07 \pm 0.23$ | $27.49 \pm 1.07$ | $33.14 \pm 1.91$ |
> > > > | CIFAR-10 | $\alpha=0.01$  | $17.82 \pm 1.14$ | $27.03 \pm 0.38$ | $28.25 \pm 2.31$ | $33.61 \pm 0.74$ |
> > > >
> > > > We note that the accuracy of FedCVAE-KD and FedCVAE-Ens in this setting is constrained by the generative capabilities of the CVAE architecture that we use, which has a simple 4 convolutional/deconvolutional-block structure. It has been shown that certain VAE architectures with alternate latent structures can generate substantially sharper images on CIFAR-10 (for example [Dai 2019]); we suspect that accuracy could be improved substantially with a more powerful architecture. However, because adopting a drastically different CVAE architecture may necessitate changes to the proposed methods, especially with respect to the knowledge distillation procedure for FedCVAE-KD, we leave further exploration of architectures to future work. Note also that the accuracy across all methods, including the baselines, is lower than what may be expected for CIFAR-10, because a simple convolutional classifier architecture is used for all methods.
> > > >
> > > > **References:**
> > > >
> > > > [Dai 2019] Dai, B. \& Wipf, D. Diagnosing and Enhancing VAE Models. Preprint at http://arxiv.org/abs/1903.05789 (2019).

---

> > > > > ### Author Response · Authors · 2022-11-16
> > > > > **Response to Reviewer TDXe (5/5)**
> > > > >
> > > > > **Q 3.7: While non-iid settings are nice to have, to me it seems that this method might suffer in less non-iid / iid settings (e.g, $\alpha = 1.0$, $\alpha = \text{inf}$); the generative distribution is more diverse and thus it would be harder to model accurately with a, hardware constrained, VAE model. This can also sort of be seen when comparing FedCVAE-KD to FedCVAE-Ens; the former has worse performance that the latter, given that the entire training distribution needs to be modelled by a single model. How does the performance of the method scale with higher values of $\alpha$?**
> > > > >
> > > > > **A 3.7:** We would like to note that real-world data often is non-IID, as stated in prior work [Karimireddy 2020, Zhu 2021, Lin 2021]. We do not believe that the IID setting is practical in real-world FL scenarios. Hence, the proposed method is designed to deal with the more challenging yet practical non-IID setting. Our focus on high statistical heterogeneity motivates the use of a generative model because, as you have alluded to in your comment, a CVAE is particularly amenable to this setting as it can directly take advantage of the simplified data distributions local to each client at higher $\alpha$.
> > > > >
> > > > > **References:**
> > > > >
> > > > > [Karimireddy 2020] Karimireddy, S. P. et al. SCAFFOLD: Stochastic Controlled Averaging for Federated Learning. In Proceedings of the 37th International Conference on Machine Learning 5132–5143 (PMLR, 2020).
> > > > >
> > > > > [Zhu 2021] Zhu, Z., Hong, J. \& Zhou, J. Data-Free Knowledge Distillation for Heterogeneous Federated Learning. In Proceedings of the 38th International Conference on Machine Learning (PMLR, 2021).
> > > > >
> > > > > [Lin 2021] Lin, T., Kong, L., Stich, S. U. \& Jaggi, M. Ensemble Distillation for Robust Model Fusion in Federated Learning. arXiv:2006.07242 [cs, stat] (2021).
> > > > >
> > > > > **Q 3.8: It seems that the evaluation against the baselines is not entirely apples-to-apples; for example, FedOneShot does not need to reveal to the server the labels and the models trained on the local data, so it has fundamentally different security/privacy tradeoffs.**
> > > > >
> > > > > **A 3.8:** We would like to clarify that in its original formulation, FedOneShot requires that clients upload their trained classifier parameters to build a model ensemble in the server [Guha 2019]. Therefore, FedOneShot generally has the same security/privacy tradeoffs as FedAvg. As for revealing client label distributions, we demonstrate our proposed methods' robustness to perturbed label distributions to promote client privacy (please see response **A 3.4** above).
> > > > >
> > > > > **References:**
> > > > >
> > > > > [Guha 2019] Guha, N., Talwalkar, A. \& Smith, V. One-Shot Federated Learning. Preprint at http://arxiv.org/abs/1902.11175 (2019).

---

> ### Comment · Reviewer_TDXe · 2022-11-18
> **Response to rebuttal**
>
> I would like to thank the authors for their extensive rebuttal. Most of my concerns have been resolved, so therefore I am increasing my score. My main concern for fully recommending acceptance is the privacy aspects; while the new experiments do help in some regard, it would be good if the authors could provide e.g., some kind of guarantee on what the noise actually hides. Now, it seems that the same scale of noise is used no matter the class which might not be able to hide the relative proportions of classes on each client (especially in this highly non-iid regime where a client mostly observes a single class). For example, if a client only has one class of data available (e.g., j), then you roughly have that $c_j \sim N(n_k, \gamma * n_k)$ whereas for other classes you have $c_i \sim N(0, \gamma * n_k)$. Now the probability of $c_j > c_i$ with $\gamma = 0.1$, is almost 1 even with $n_k$ is as small as 10.

---

> > ### Author Response · Authors · 2022-11-19
> > **Follow-up to Reviewer TDXe**
> >
> > First of all, thank you very much for increasing your score; we appreciate your diligence and constructive criticism during this review process.
> >
> > In our approach using Gaussian noise to mask the label distributions, the goal is not necessarily to *fully* mask the relative proportions, but to instead inspect our proposed methods' sensitivity towards reduced label distribution information. Indeed, it seems that it would not be necessary to fully mask label distributions to promote privacy in this regard. As you point out, as $\alpha \to 0$ it may be nearly impossible to properly mask the relative proportions since there exists such a disparity between class counts; perhaps some method that sets the noise variance in inverse proportion to $\alpha$ could help remedy this (e.g., set $\gamma \propto \frac{1}{\alpha}$). We would also like to point out that while in the manuscript we state that class counts should be uploaded to be consistent with prior work [Zhu 2021, Zhang 2022], both of our proposed methods only require the label proportions from each client for sampling conditioning classes $\mathbf{y}$, which is strictly less informative than full counts.
> >
> > As for guarantees on the information hidden by adding noise: the notion of uploading and harnessing client label distributions in FL is quite new (the papers we follow in this regard are from July 2021 and June 2022) and quantifying the privacy risks that label distributions might induce is very much an open problem which we believe could benefit from focused development. We have added an acknowledgment of the importance of this analysis as a direction for future theoretical development in the the revised manuscript (please see Appendix C, "Adding Noise to Uploaded Label Distributions").
> >
> > **References:**
> >
> > [Zhu 2021] Zhu, Z., Hong, J. \& Zhou, J. Data-Free Knowledge Distillation for Heterogeneous Federated Learning. In Proceedings of the 38th International Conference on Machine Learning (PMLR, 2021).
> >
> > [Zhang 2022] Zhang, L., Shen, L., Ding, L., Tao, D. \& Duan, L.-Y. Fine-Tuning Global Model via Data-Free Knowledge Distillation for Non-IID Federated Learning. In Proceedings of the IEEE/CVF Conference on Computer Vision and Pattern Recognition (CVPR, 2022).}

---

### Official Review · Reviewer_mEpo · 2022-11-03

**Confidence:** 3
**Correctness:** 3
**Technical Novelty And Significance:** 2
**Empirical Novelty And Significance:** 3
**Recommendation:** 6

**Clarity, Quality, Novelty And Reproducibility:**

The paper is clear and well-written, and the algorithms are described precisely enough for the results to be reproducible. The novelty of the paper and its contributions are marginal, in my opinion.

**Strength And Weaknesses:**

The paper is very well written and easy to follow. The numerical experiments are well executed and show promising performance achieved by the proposed methods.

That being said, there are some downsides that, in my opinion, slightly outweigh the strengths of the paper. On the problem formulation side, I believe the one-shot federated learning (FL) problem is not very well motivated; the point of FL is to enable training models at different clients by leveraging each others' data through communication without explicitly sharing them. In a one-shot scenario, the problem essentially reduces to having a set of models, which need to be aggregated into a single, universal model, so I am still not convinced how the "federated" aspect of the problem is of the essence.

On the contributions side, the main issue I have with the paper is a lack of a central, key novelty. In its current form, the paper is mostly a collection of different ideas, namely two algorithms that are almost independent of each other, as well as a security aspect, which is again mostly orthogonal to the proposed methods. To me, what would make this paper much stronger is to focus on only one of these algorithms/aspects throughout the paper. For example, since the performance of FEDCVAE-KD is better than baselines, I believe it would be better to focus on this algorithm in the main body of the paper, and add the ensemble version of the algorithm (which, as the authors allude to, requires a higher amount of memory that scales with the number of clients) as an extension, in a separate discussion section toward the end of the paper and/or in the appendix. Moreover, the security and privacy considerations are definitely important, but including them in the main body of the paper stretches it too thin, in my opinion.

Further comments:

- Aside from the solid empirical performance, are there any intuitive/theoretical explanations as to why the proposed methods are well suited for i) *one-shot* FL, and ii) *high heterogeneity* FL? As presented, I cannot see a clear motivation for why the proposed methods should perform better in the considered problem formulation.
- Please discuss whether or not sharing local label distributions with the server violates the privacy of clients' data. How do the proposed methods perform if this information is unavailable at the server?
- Presenting the flowchart figure for only one of the proposed methods (FEDCVAE-KD) in Figure 2 and not including any of the algorithms for the two proposed methods makes the paper slightly incoherent, without switching back and forth between the body of the paper and the appendix. I suggest rearranging the contents to remedy this issue as best as possible by at least having both the figure and algorithm for one of the methods or having the figures for both methods in the main body of the paper.
- In the first footnote on page 4, why is a uniform distribution used for sampling $\mathbf{z}$? Why not, for example, use a truncated Gaussian distribution?
- In my opinion, including FedAVG is slightly unfair in the experiments. Under FedAVG, a set of client model parameters $\\{\mathbf{w}^k\\}_{k=1}^m$ are aggregated using a simple average, i.e., $\mathbf{w}=\frac{1}{m} \sum_\{k=1\}^m\mathbf{w}^k$. In a very heterogeneous case, each of these model parameters is the minimizer of a very different objective function over an entirely different optimization landscape, so it is completely expected that a linear averaging of these model parameters should lead to inferior performance, especially if the averaging operation is done only once.
- Following the above comment, I believe your primary competing baseline is FedONESHOT, so I suggest adding more details on this algorithm's main idea.
- Do you have any explanation as to why the standard deviation of the performance of the proposed methods is, on average, larger under SVHN in Table 2?

**Summary Of The Paper:**

This paper introduces two methods, FEDCVAE-ENS and FEDCVAE-KD, for one-shot federated learning with extreme data heterogeneity among clients. Once client models are trained using conditional variational auto-encoder (CVAE) architectures, the decoders, alongside label distributions of all clients, are uploaded to the server. FEDCVAE-ENS creates an ensemble of synthetic data samples created via all client decoders to create a dataset that can be used to train a centralized classifier at the server. FEDCVAE-KD, on the contrary, trains a centralized decoder at the server using knowledge distilled from the client decoders. The trained decoder is then used to create a dataset of synthetic samples, which is used to train the centralized classifier. Simulation results show that i ) the proposed methods considerably outperform baselines when the data samples are very heterogeneous, and ii) using the CVAE architecture enables a security-promoting extension of the proposed methods that prevents an eavesdropper from training a well-performing centralized classifier.

**Summary Of The Review:**

I think this is a borderline paper with clear positive aspects, but one that needs some content streamlining and restructuring, as well as stronger motivation so that the contributions are better appreciated.

---

> ### Author Response · Authors · 2022-11-16
> **Response to Reviewer mEpo (1/5)**
>
> **Q 2.1: On the problem formulation side, I believe the one-shot federated learning (FL) problem is not very well motivated; the point of FL is to enable training models at different clients by leveraging each others' data through communication without explicitly sharing them. In a one-shot scenario, the problem essentially reduces to having a set of models, which need to be aggregated into a single, universal model, so I am still not convinced how the "federated" aspect of the problem is of the essence.**
>
> **A 2.1:** One-shot federated learning (FL) certainly differs from standard FL in some ways, but it sits firmly within the "federated" framework. Like FL, one-shot FL does not share the data explicitly with the server; in our method, we instead upload client decoders to generate synthetic data for model training, which is similar to the procedure in [Zhu 2021]. Furthermore, once the one-shot FL procedure is complete, additional communication from server to client can occur: for instance, in the cross-silo medical setting, the resulting server classifier could be communicated back to clients for use in clinical applications. In essence, one-shot FL is a very strict subset of standard FL; instead of reducing the number of communication rounds, we squeeze as much learning as possible into the first round. But this does not necessarily make it incompatible with standard FL. In fact, some papers have explored using the output of a one-shot procedure as the model initialization for multiple-round FL [Li 2021], highlighting an additional connection between the two settings.
>
> **References:**
>
> [Zhu 2021] Zhu, Z., Hong, J. \& Zhou, J. Data-Free Knowledge Distillation for Heterogeneous Federated Learning. In Proceedings of the 38th International Conference on Machine Learning (PMLR, 2021).
>
> [Li 2021] Li, Q., He, B. \& Song, D. Practical One-Shot Federated Learning for Cross-Silo Setting. in Proceedings of the Thirtieth International Joint Conference on Artificial Intelligence 1484–1490 (IJCAI, 2021). doi:10.24963/ijcai.2021/205.
>
> **Q 2.2: The main issue I have with the paper is a lack of a central, key novelty. In its current form, the paper is mostly a collection of different ideas, namely two algorithms that are almost independent of each other, as well as a security aspect, which is again mostly orthogonal to the proposed methods. To me, what would make this paper much stronger is to focus on only one of these algorithms/aspects throughout the paper. For example, since the performance of FedCVAE-KD is better than baselines, I believe it would be better to focus on this algorithm in the main body of the paper, and add the ensemble version of the algorithm (which, as the authors allude to, requires a higher amount of memory that scales with the number of clients) as an extension, in a separate discussion section toward the end of the paper and/or in the appendix. Moreover, the security and privacy considerations are definitely important, but including them in the main body of the paper stretches it too thin, in my opinion.**
>
> **A 2.2:** Thank you for your suggestions. To streamline the methods description in Section 3, we have moved the details of FedCVAE-Ens to Appendix A, because FedCVAE-KD is a memory-efficient extension of FedCVAE-Ens. We have also moved the discussion of the privacy-promoting extension using differential privacy to Appendix B, but leave a short description of the security-promoting extension in the main text since we believe it represents a non-invasive and empirically effective method for promoting pipeline security. With these changes, the manuscript is more streamlined.

---

> > ### Author Response · Authors · 2022-11-16
> > **Response to Reviewer mEpo (2/5)**
> >
> > **Q 2.3: Aside from the solid empirical performance, are there any intuitive/theoretical explanations as to why the proposed methods are well suited for i) *one-shot* FL, and ii) *high heterogeneity* FL? As presented, I cannot see a clear motivation for why the proposed methods should perform better in the considered problem formulation.**
> >
> > **A 2.3:** Intuitively, our proposed methods are suitable for the high statistical heterogeneity setting precisely because of the simplified conditional data distributions that arise, which CVAEs can easily capture. Figure 1 in the manuscript shows how client decoders become experts in the few classes that they observed, visually demonstrating an intuitive argument for why the proposed methods are well-suited for high heterogeneity FL. We have additional motivating text demonstrating the suitability of our proposed methods in Section 1 and Figure 1 of the manuscript.
> >
> > At the intersection of high statistical heterogeneity and one-shot FL, we have many clients that each have very narrow local data distributions and must extract an understanding of the global data distribution in one round. Approaches based on discriminative models (e.g., FedAvg or FedOneShot) collapse without a global view of the data distribution, which requires iterative communication over multiple rounds. Our proposed methods focus on directly capturing these narrow conditional distributions in a generative sense, which does not necessarily benefit from an understanding of the global data distribution since more diverse distributions are more difficult to capture.

---

> > > ### Author Response · Authors · 2022-11-16
> > > **Response to Reviewer mEpo (3/5)**
> > >
> > > **Q 2.4: Please discuss whether or not sharing local label distributions with the server violates the privacy of clients' data. How do the proposed methods perform if this information is unavailable at the server?**
> > >
> > > **A 2.4:** Multiple well-cited works have used label distribution communication as a crucial element of their proposed methods: both [Zhu 2021] and [Zhang 2022] require communication of label distributions and do not bring up major concerns over violations of client privacy.
> > >
> > > As requested, we have conducted additional experiments to elicit the relationship between the server classifier accuracy and the amount of information included in uploaded label distributions $\hat{p}^k(\mathbf{y})$. We follow [Zhang 2022] and add noise to the uploaded label distributions. We draw noise from a normal distribution $\epsilon_c \sim \mathcal{N}(0, \gamma \cdot n_k)$ such that the ``strength'' (variance) of the noise applied to class $c$ is in proportion $\gamma$ to the total number of training samples for client $k$. Noise is applied to the training sample count for each class for a given client before uploading this information to the server. As $\gamma \to \infty$, information from the uploaded label distribution disappears. When $\gamma = 0$, the exact local label distributions are communicated. Figure 9 in the revised manuscript visually shows how adding noise affects the label distributions. We also test the case where no local label distribution information is uploaded to the server, i.e., we uniformly sample conditioning classes when extracting synthetic samples from client decoders.
> > >
> > > The following tables include new experimental results for adding varying amounts of noise to the uploaded label distributions (Table 8 in the manuscript). The first of the tables below shows that when adding a modest amount of noise to the client label distributions (i.e., $\gamma \le 0.1$), accuracy for both FedCVAE-Ens and FedCVAE-KD is barely affected compared to uploading with no noise. Both proposed methods with varying amounts of noise continue to beat the baselines (compare to results in Table 1 of the manuscript where $\alpha = 0.01$). In the second table below, we display results of not uploading client label distributions to the server. Instead of using client label distributions for sampling conditioning classes $\mathbf{y}$, we use a uniform distribution. As expected, performance suffers, because the samples will be incorrectly labeled for the classes that the client decoder observed less, misleading the server classifier. However, we reiterate that small noise perturbations to the label distributions could offer improved privacy, and methods from previous work also rely on uploading client label distributions to the server.
> > >
> > > **Table:** Results for adding noise to the uploaded client label distributions
> > > | Dataset      | Noise Proportion |    FedCVAE-KD    |    FedCVAE-Ens   |
> > > |--------------|------------------|:----------------:|:----------------:|
> > > | MNIST        | $\gamma=0.01$    | $81.10 \pm 0.69$ | $93.69 \pm 0.82$ |
> > > | MNIST        | $\gamma=0.05$    | $80.41 \pm 1.81$ | $92.65 \pm 0.77$ |
> > > | MNIST        | $\gamma=0.1$     | $80.56 \pm 0.36$ | $92.65 \pm 1.33$ |
> > > | FashionMNIST | $\gamma=0.01$    | $69.43 \pm 1.61$ | $76.44 \pm 1.11$ |
> > > | FashionMNIST | $\gamma=0.05$    | $68.57 \pm 1.74$ | $74.44 \pm 2.14$ |
> > > | FashionMNIST | $\gamma=0.1$     | $69.07 \pm 1.97$ | $72.88 \pm 3.03$ |
> > > | SVHN         | $\gamma=0.01$    | $57.18 \pm 2.24$ | $64.03 \pm 1.78$ |
> > > | SVHN         | $\gamma=0.05$    | $56.93 \pm 2.43$ | $62.68 \pm 2.64$ |
> > > | SVHN         | $\gamma=0.1$     | $55.85 \pm 2.29$ | $61.47 \pm 1.83$ |
> > > | | |
> > >
> > >
> > > **Table:** Results for our proposed methods when client label distributions are not uploaded, i.e., uniform sampling across classes.
> > > | Dataset      | FedCVAE-KD       | FedCVAE-Ens      |
> > > |--------------|------------------|------------------|
> > > | MNIST        | $55.51 \pm 3.26$ | $60.71 \pm 2.59$ |
> > > | FashionMNIST | $27.84 \pm 4.85$ | $41.66 \pm 5.15$ |
> > > | SVHN         | $11.00 \pm 2.68$ | $18.09 \pm 6.27$ |
> > > |       |     |
> > >
> > > **References:**
> > >
> > > [Zhu 2021] Zhu, Z., Hong, J. \& Zhou, J. Data-Free Knowledge Distillation for Heterogeneous Federated Learning. In Proceedings of the 38th International Conference on Machine Learning (PMLR, 2021).
> > >
> > > [Zhang 2022] Zhang, L., Shen, L., Ding, L., Tao, D. \& Duan, L.-Y. Fine-Tuning Global Model via Data-Free Knowledge Distillation for Non-IID Federated Learning. In Proceedings of the IEEE/CVF Conference on Computer Vision and Pattern Recognition (CVPR, 2022).

---

> > > > ### Author Response · Authors · 2022-11-16
> > > > **Response to Reviewer mEpo (4/5)**
> > > >
> > > > **Q 2.5: Presenting the flowchart figure for only one of the proposed methods (FEDCVAE-KD) in Figure 2 and not including any of the algorithms for the two proposed methods makes the paper slightly incoherent, without switching back and forth between the body of the paper and the appendix. I suggest rearranging the contents to remedy this issue as best as possible by at least having both the figure and algorithm for one of the methods or having the figures for both methods in the main body of the paper.**
> > > >
> > > > **A 2.5:** As suggested, we moved the algorithm for FedCVAE-KD from the appendix to the main body of the manuscript. In addition, we have moved the description of the other method, FedCVAE-Ens, to Appendix A.
> > > >
> > > > **Q 2.6: In the first footnote on page 4, why is a uniform distribution used for sampling $\mathbf{z}$? Why not, for example, use a truncated Gaussian distribution?**
> > > >
> > > > **A 2.6:** A tight uniform distribution over the highest density region of the normal prior was used simply because of improved empirical results during initial experiments. As suggested, we have conducted additional experiments using a standard normal distribution truncated to several bounds (i.e., $\pm 1$, $\pm 2$, and $\pm 3$ standard deviations) for sampling latent vectors. Qualitatively, we observed more diverse samples from the decoders when using the truncated normal compared to a uniform distribution, likely because of the closer match with the actual prior. This was reflected in a modest to substantial increase in accuracy across many of the experimental results. We have updated the pertinent figures and tables accordingly with the improved results in our revised version.
> > > >
> > > > Table 1 (also shown below) in the manuscript shows the performance of all models across varying levels of statistical heterogeneity using a truncated normal distribution as the prior. We note that FedCVAE-Ens now obtains over 95\% accuracy at $\alpha = 0.001$ on MNIST, and the best baseline method, FedAvg, achieves just 45.12\% accuracy. The greatest improvements can be seen for the dataset SVHN, with roughly a 10\% improvement in accuracy across all levels of $\alpha$ for FedCVAE-Ens and 6-7\% improvement for FedCVAE-KD. The results across the number of clients (Table 2 and Table 5 in the manuscript), heterogeneous local models (Figure 4 in the manuscript), various aggregation methods for FedCVAE-KD (Table 6 in the manuscript), and different percentage subsets of the data (Table 7 in the manuscript) have also been updated to reflect the use of a truncated normal distribution for decoder sampling. For more details, please refer to the revised manuscript.
> > > >
> > > > **Table:** Results for all methods using a truncated standard normal distribution for sampling from decoders for both of our proposed methods.
> > > > | Dataset      | Heterogeneity  |      FedAvg      |    FedOneShot    |    FedCVAE-KD    |    FedCVAE-Ens   |
> > > > |--------------|----------------|:----------------:|:----------------:|:----------------:|:----------------:|
> > > > | MNIST        | $\alpha=0.001$ | $45.12 \pm 5.87$ | $11.90 \pm 0.40$ | $82.24 \pm 1.09$ | $95.37 \pm 0.52$ |
> > > > | MNIST        | $\alpha=0.01$  | $58.29 \pm 2.83$ | $41.15 \pm 0.70$ | $82.01 \pm 1.61$ | $93.83 \pm 1.53$ |
> > > > | MNIST        | $\alpha=0.05$  | $80.10 \pm 2.35$ | $82.95 \pm 0.49$ | $79.57 \pm 1.17$ | $91.86 \pm 0.75$ |
> > > > | FashionMNIST | $\alpha=0.001$ | $32.77 \pm 4.52$ | $10.00 \pm 0.00$ | $69.53 \pm 1.70$ | $76.04 \pm 0.93$ |
> > > > | FashionMNIST | $\alpha=0.01$  | $45.85 \pm 2.95$ | $37.63 \pm 0.53$ | $69.97 \pm 1.63$ | $76.62 \pm 1.61$ |
> > > > | FashionMNIST | $\alpha=0.05$  | $46.86 \pm 2.37$ | $64.53 \pm 1.73$ | $67.24 \pm 1.96$ | $71.95 \pm 2.17$ |
> > > > | SVHN         | $\alpha=0.001$ | $20.31 \pm 4.36$ | $15.94 \pm 0.00$ | $55.48 \pm 2.12$ | $65.52 \pm 0.66$ |
> > > > | SVHN         | $\alpha=0.01$  | $25.12 \pm 2.07$ | $31.91 \pm 1.26$ | $55.97 \pm 0.38$ | $65.50 \pm 2.28$ |
> > > > | SVHN         | $\alpha=0.05$  | $33.66 \pm 4.12$ | $49.60 \pm 1.59$ | $54.24 \pm 2.08$ | $66.61 \pm 2.09$ |
> > > > | | |

---

> > > > > ### Author Response · Authors · 2022-11-16
> > > > > **Response to Reviewer mEpo (5/5)**
> > > > >
> > > > > **Q 2.7: In my opinion, including FedAvg is slightly unfair in the experiments. Under FedAvg, a set of client model parameters $ \\{ \mathbf{w}^k \\}_ {k=1}^{m} $ are aggregated using a simple average, i.e., $\mathbf{w} = \frac{1}{m} \sum_{k=1}^{m} \mathbf{w}^k$. In a very heterogeneous case, each of these model parameters is the minimizer of a very different objective function over an entirely different optimization landscape, so it is completely expected that a linear averaging of these model parameters should lead to inferior performance, especially if the averaging operation is done only once.**
> > > > >
> > > > > **A 2.7:** We agree that FedAvg is expected to perform poorly under heterogeneous client data partitions given the large differences in optimization landscape, generating incompatible parameters for averaging. However, in the absence of one-shot FL methods that are comparable to our proposed methods (i.e., are both suitable for single-round FL and do not rely on auxiliary data), as we cover in detail in Section 4.1, FedAvg represents one of the only available baselines. Additionally, as in the larger FL literature, FedAvg is a standard baseline method in the one-shot FL literature [Zhang 2021, Shin 2020, Li 2021, Zhou 2021]. For these reasons, we believe the inclusion of FedAvg as a baseline method is warranted. We also compare our method to FedOneShot as a more competitive baseline.
> > > > >
> > > > > **References:**
> > > > >
> > > > > [Zhang 2021] Zhang, J. et al. A Practical Data-Free Approach to One-shot Federated Learning with Heterogeneity. Preprint at http://arxiv.org/abs/2112.12371 (2021).
> > > > >
> > > > > [Shin 2020] Shin, M. et al. XOR Mixup: Privacy-Preserving Data Augmentation for One-Shot Federated Learning. Preprint at https://doi.org/10.48550/arXiv.2006.05148 (2020).
> > > > >
> > > > > [Li 2021] Li, Q., He, B. \& Song, D. Practical One-Shot Federated Learning for Cross-Silo Setting. In Proceedings of the Thirtieth International Joint Conference on Artificial Intelligence 1484–1490 (IJCAI, 2021). doi:10.24963/ijcai.2021/205.
> > > > >
> > > > > [Zhou 2021] Zhou, Y., Pu, G., Ma, X., Li, X. \& Wu, D. Distilled One-Shot Federated Learning. Preprint at https://doi.org/10.48550/arXiv.2009.07999 (2021).
> > > > >
> > > > > **Q 2.8: Following the above comment, I believe your primary competing baseline is FedOneShot, so I suggest adding more details on this algorithm's main idea.**
> > > > >
> > > > > **A 2.8:** We added more details in Section 4.1 of the manuscript. Specifically, in the main text we state, "FedOneShot ensembles the predictions of select uploaded client classifiers using a sampling procedure; because we consider substantially less clients than Guha et al. (2019), we disregard sampling and use all clients in the ensemble."
> > > > >
> > > > > **Q 2.9: Do you have any explanation as to why the standard deviation of the performance of the proposed methods is, on average, larger under SVHN in Table 2?**
> > > > >
> > > > > **A 2.9:** In Table 2 of the manuscript, which has been updated to use the results from the truncated standard normal sampling, FedCVAE-Ens has standard deviations of $0.86-2.16$ across MNIST and FasionMNIST, and $1.89-2.44$ for SVHN; for FedCVAE-KD, standard deviations range from $1.13-1.76$ across MNIST and FasionMNIST, and $0.38-2.75$ for SVHN. The range of standard deviations are overlapping when comparing SVHN to MNIST/FashionMNIST in these new results, so we do not think that there is cause for concern in this regard.

---

> ### Author Response · Authors · 2022-11-18
> **Thanks for your comments and please look at our response**
>
> Dear Reviewer mEpo,
>
> We would like to thank you for taking the time to review our paper and for the insightful comments.
>
> We have addressed all the comments and suggestions you made. In particular, as suggested, we have moved the details of FedCVAE-Ens to Appendix A in order to focus on the main contributions of our work. We have also moved the discussion of the privacy-promoting extension using differential privacy to Appendix B. In addition, we have conducted additional experiments using a truncated Gaussian distribution for sampling $\mathbf{z}$.
>
> Please kindly let us know if you have any additional concerns. We truly appreciate this opportunity to improve our work and shall be most grateful for any feedback you could give to us.

---

> > ### Author Response · Authors · 2022-11-29
> > **Thank you for your feedback and reminder to view our rebuttal**
> >
> > Reviewer mEpo,
> >
> > We would like to gently emphasize the comment above. Please let us know if you have any lingering concerns so that we can address them as soon as possible. Thank you!

---

> > ### Comment · Reviewer_mEpo · 2022-12-02
> > **Thanks for the rebuttal!**
> >
> > Thank you very much for the time and effort in preparing the rebuttal and the revised manuscript. Several concerns of mine have been addressed, and therefore, I have raised my score.

---

### Official Review · Reviewer_nh59 · 2022-11-03

**Confidence:** 4
**Correctness:** 4
**Technical Novelty And Significance:** 3
**Empirical Novelty And Significance:** Not applicable
**Recommendation:** 6

**Clarity, Quality, Novelty And Reproducibility:**

The overall clarity and quality of the paper is good. The method proposed is novel. Details for reproducibility are provided in the paper.

**Strength And Weaknesses:**

### Strength

* The proposed method of one-shot FL is very interesting and novel.
* The proposed method doesn't require auxiliary public dataset.
* Both security and privacy extensions are considered in the paper.
* It can be applied to heterogeneous local models with similar generative capabilities.

### Weaknesses
* One potential issue of the method is training a CVAE locally. In the paper, the images used in the dataset are very small, and the amount of data each client has is large. However, in the real-world application of FL like medical imagings, with very large images and very limited number of data samples, training a CVAE locally can be difficult even just for one class of data.
* The training task of local models is different from the original task. This limits the re-use of pre-trained local model from original task. I think one of the reasons that people want to do one-shot FL is to prevent any re-training locally. But the proposed method still requires to train some new model.

**Summary Of The Paper:**

This paper considers one-shot federated learning under high statistical heterogeneity. They propose a data-free FL mothods FEDCVAE-ENS and its extension FEDCVAEKD. Conditional variantional autoencoders(CVAE) are trained locally and then the decoders are sent to the server. At the server, either ensemble or knowledge distillation is used to generate an ensemble dataset which is later used to train the final model. The paper conducts experiments on three datasets to demenstrate its effectiveness under high statistical heterogeneity.

**Summary Of The Review:**

The idea of the paper is novel and interesting. Although in real applications with larger images and limited data samples, it may not work very well, it provides a novel perspective of doing one-shot FL.

---

> ### Author Response · Authors · 2022-11-16
> **Response to Reviewer nh59 (1/2)**
>
> **Q 1.1: One potential issue of the method is training a CVAE locally. In the paper, the images used in the dataset are very small, and the amount of data each client has is large. However, in the real-world application of FL like medical images, with very large images and very limited number of data samples, training a CVAE locally can be difficult even just for one class of data.**
>
> **A 1.1:** To better examine the relationship between client dataset size and performance in the context of our study, we have conducted additional experiments under different percent subsets of the training data when $\alpha=0.01$ and $m=10$, as shown in the following table. We can observe that with only 10\% of the available MNIST and FashionMNIST training data, FedCVAE-Ens and FedCVAE-KD still achieve comparable accuracy to that of 50\% of training data. If the percentage of data is further reduced to 5\%, while accuracy for both FedCVAE-Ens and FedCVAE-KD decreases, they more than double the accuracy of FedAvg and FedOneShot for FashionMNIST and SVHN. Both proposed methods also vastly outperform the baselines with 5\% of the data for MNIST. Hence, we can conclude that the proposed methods would not collapse under too little data and do much better than baseline methods. We agree that training a CVAE locally could be difficult in the case of clients with few data samples or very large images. However, the benchmarks chosen are standard in both the FL and one-shot FL literature [Zhang 2021, Zhou 2021, Li 2021, Shin 2020]. In terms of image size, there exist proposed VAE variants that scale better to larger image sizes; [Vahdat 2021] is a highly-cited recent development in this area.
>
> **Table:** Results for all methods varying the number of samples for each client by reducing the portion of the training set distributed to clients.
> | Dataset      | \% Subset |      FedAvg      |    FedOneShot    |    FedCVAE-KD    |    FedCVAE-Ens   |
> |--------------|-----------|:----------------:|:----------------:|:----------------:|:----------------:|
> | MNIST        | 5\%       | $40.65 \pm 2.65$ | $34.24 \pm 1.51$ | $74.98 \pm 4.82$ | $76.47 \pm 2.58$ |
> | MNIST        | 10\%      | $38.55 \pm 4.43$ | $36.86 \pm 1.14$ | $81.24 \pm 1.45$ | $89.95 \pm 1.27$ |
> | MNIST        | 25\%      | $56.17 \pm 3.82$ | $47.50 \pm 0.72$ | $80.00 \pm 1.52$ | $92.41 \pm 0.79$ |
> | MNIST        | 50\%      | $58.29 \pm 2.83$ | $41.15 \pm 0.70$ | $82.01 \pm 1.61$ | $93.83 \pm 1.53$ |
> | FashionMNIST | 5\%       | $14.11 \pm 2.33$ | $26.96 \pm 0.74$ | $66.75 \pm 1.80$ | $69.78 \pm 1.26$ |
> | FashionMNIST | 10\%      | $26.53 \pm 6.28$ | $27.26 \pm 1.69$ | $69.07 \pm 1.53$ | $72.10 \pm 1.66$ |
> | FashionMNIST | 25\%      | $28.29 \pm 1.69$ | $29.97 \pm 2.53$ | $71.24 \pm 1.48$ | $75.31 \pm 1.42$ |
> | FashionMNIST | 50\%      | $45.85 \pm 2.95$ | $37.63 \pm 0.53$ | $69.97 \pm 1.63$ | $76.62 \pm 1.61$ |
> | SVHN         | 5\%       | $17.19 \pm 5.38$ | $11.08 \pm 0.38$ | $41.74 \pm 3.67$ | $46.06 \pm 1.29$ |
> | SVHN         | 10\%      | $22.97 \pm 4.37$ | $16.16 \pm 0.73$ | $47.65 \pm 2.88$ | $54.36 \pm 1.76$ |
> | SVHN         | 25\%      | $28.40 \pm 3.15$ | $29.10 \pm 0.26$ | $52.15 \pm 1.23$ | $59.96 \pm 2.68$ |
> | SVHN         | 50\%      | $20.38 \pm 1.03$ | $32.39 \pm 2.85$ | $57.56 \pm 1.09$ | $64.82 \pm 1.27$ |
> | SVHN         | 100\%     | $25.12 \pm 2.07$ | $31.91 \pm 1.26$ | $55.97 \pm 0.38$ | $65.50 \pm 0.28$ |
> |                   |                     |                          |                                  |                         |               |
>
> **References:**
>
> [Zhang 2021] Zhang, J. et al. A Practical Data-Free Approach to One-shot Federated Learning with Heterogeneity. Preprint at http://arxiv.org/abs/2112.12371 (2021).
>
> [Zhou 2021] Zhou, Y., Pu, G., Ma, X., Li, X. \& Wu, D. Distilled One-Shot Federated Learning. Preprint at https://doi.org/10.48550/arXiv.2009.07999 (2021).
>
> [Li 2021] Li, Q., He, B. \& Song, D. Practical One-Shot Federated Learning for Cross-Silo Setting. in Proceedings of the Thirtieth International Joint Conference on Artificial Intelligence 1484–1490 (IJCAI, 2021). doi:10.24963/ijcai.2021/205.
>
> [Shin 2020] Shin, M. et al. XOR Mixup: Privacy-Preserving Data Augmentation for One-Shot Federated Learning. Preprint at https://doi.org/10.48550/arXiv.2006.05148 (2020).
>
> [Vahdat 2021] Vahdat, A. \& Kautz, J. NVAE: A Deep Hierarchical Variational Autoencoder. arXiv:2007.03898 [cs, stat] (2021).

---

> > ### Author Response · Authors · 2022-11-16
> > **Response to Reviewer nh59 (2/2)**
> >
> > **Q 1.2: The training task of local models is different from the original task. This limits the re-use of pre-trained local model from original task. I think one of the reasons that people want to do one-shot FL is to prevent any re-training locally. But the proposed method still requires to train some new model.**
> >
> > **A 1.2:** We would like to note that, according to the prior work [Guha 2019, Li 2021], the standard goal of one-shot federated learning (FL) is to assemble a performant model in the server. In the context of our paper, the server classifier parameters $\mathbf{w}^S_C$, which should perform relatively well when applied to the global data distribution, could be communicated to clients after the FL process is complete if this is the FL agreement of the constituent clients. If further model personalization is the client's goal, then additional training/fine-tuning may be needed, but the general aim of one-shot FL does not necessarily accommodate personalization.
> >
> > **References:**
> >
> > [Guha 2019] Guha, N., Talwalkar, A. \& Smith, V. One-Shot Federated Learning. Preprint at http://arxiv.org/abs/1902.11175 (2019).
> >
> > [Li 2021] Li, Q., He, B. \& Song, D. Practical One-Shot Federated Learning for Cross-Silo Setting. in Proceedings of the Thirtieth International Joint Conference on Artificial Intelligence 1484–1490 (IJCAI, 2021). doi:10.24963/ijcai.2021/205.

---

> ### Author Response · Authors · 2022-11-29
> **Thank you for your feedback and reminder to view our rebuttal**
>
> Reviewer nh59,
>
> We appreciate your insightful comments and feedback on the initial version of the manuscript. We have revised the manuscript according to your comments, in particular adding experiments to test accuracy with less data samples per client. If you have any additional concerns, please let us know so we can address them as soon as possible. Thank you!

---

> ### Comment · Reviewer_nh59 · 2022-12-13
> **post-rebuttal response**
>
> Thank you for the additional experiments with subset of the data. However the reviewer's concerns remain. Therefore, I will keep my score.
>
> - 5% data with small size image like MNIST and FashionMNIST still cannot justify the scalability of the method on real world applications with larger image.
> - The proposed method is unable to re-use any preivously pre-trained local model with target task.

---

### Author Response · Authors · 2022-11-16
**General Response to Reviewers**

We thank the reviewers for their constructive comments and thorough reviews. We have responded to each reviewer's comments below and have uploaded a revised version of the manuscript. Please let us know if you have any additional concerns.

---

### Decision · Program_Chairs · 2023-01-20

**Decision:**

Accept: poster

**Justification For Why Not Higher Score:**

all scores are marginal accept so does not merit a higher score

**Justification For Why Not Lower Score:**

all reviewers are inagreement

**Metareview: Summary, Strengths And Weaknesses:**

Federated learning (FL) is a distributed learning framework that collaboratively trains a shared model without transferring local clients' data to a central server. One-shot FL limits communication to a single round while attempting to retain performance. However, performance often degrades under high statistical heterogeneity, fails to promote pipeline security, or requires an auxiliary public dataset. To address these limitations, two novel data-free one-shot FL methods are proposed: FedCVAE-Ens and its extension FedCVAE-KD. These approaches reframe the local learning task using a conditional variational autencoder (CVAE) to address high statistical heterogeneity. In addition, FedCVAE-KD uses knowledge distillation to compress the ensemble of client decoders into a single decoder. A method that shifts the center of the CVAE prior distribution is also proposed to promote security. The authors claim that these methods can incorporate heterogeneous local models, and they carryout various experiments using multiple benchmark datasets to confirm this. The reviewers thought the one-shot FL was interesting and novel. They also liked the fact that the proposed method does not require auxiliary public datasets and that both security and privacy was considered and that it can be applied to heterogeneous local models. The reviewers raised issues about training CVAE locally and the fact that the training task of local models is different from the original task, simplicity of the datasets, some of the comparisons in the numerical experiments and lack of central novelty. The authors provided a thorough response that mitigated many of the concerns. My own reading is that this is an interesting paper and therefore recommend acceptance. I do recommend following the great suggestions of the reviewers for the final version.



**Note From Pc:**

if the above contains the word "oral" or "spotlight" please see: "oral" presentation means -> notable-top-5% and "spotlight" means -> notable-top-25%. As stated in our emails, we are disassociating presentation type from AC recommendations